# AllSim: Simulating and Benchmarking Resource Allocation Policies in Multi-User Systems

**Jeroen Berrevoets**
University of Cambridge

**Daniel Jarrett**
University of Cambridge

**Alex J. Chan**
University of Cambridge

**Mihaela van der Schaar**
University of Cambridge

## Abstract

Numerous real-world systems, ranging from healthcare to energy grids, involve users competing for finite and potentially scarce resources. Designing policies for repeated resource allocation in such real-world systems is challenging for many reasons, including the changing nature of user types and their (possibly urgent) need for resources. Researchers have developed numerous machine learning solutions for determining repeated resource allocation policies in these challenging settings. However, a key limitation has been the absence of good methods and test-beds for benchmarking these policies; almost all resource allocation policies are benchmarked in environments which are either completely synthetic or do not allow *any* deviation from historical data. In this paper we introduce AllSim, which is a benchmarking environment for realistically simulating the impact and utility of policies for resource allocation in systems in which users compete for such scarce resources. Building such a benchmarking environment is challenging because it needs to successfully take into account *the entire collective* of potential users and the impact a resource allocation policy has on all the other users in the system. AllSim's benchmarking environment is modular (each component being parameterized individually), learnable (informed by historical data), and customizable (adaptable to changing conditions). These, when interacting with an allocation policy, produce a dataset of simulated outcomes for evaluation and comparison of such policies. We believe AllSim is an essential step towards a more systematic evaluation of policies for scarce resource allocation compared to current approaches for benchmarking such methods.

## 1 Introduction

The problem of repeated resource allocation to users with timeliness constraints is ubiquitous in settings ranging from healthcare to engineering systems and even labour markets. This problem becomes even more challenging when the resources are diverse and users may derive different benefits from obtaining a specific resource. In these applications, a resource coordinator or decision maker, is tasked with allocating these diverse resources to a pool of diverse users which arrive or leave over time.

When a new resource arrives, it is the decision maker's task to assign this coveted resource to one of the users in their current pool. Making such a decision has an enormous impact on the *complete* system: (i) when a resource is allocated to a user, other users now need to wait for the next resource to arrive, thereby impacting their utility (since they are delay-sensitive), (ii) the match between the resource and the user recipient has different valuations, (iii) assigning a resource to a specific user in the pool influences the utilities of *all* the other users in the pool as well, and thereby impacting any subsequent

37th Conference on Neural Information Processing Systems (NeurIPS 2023) Track on Datasets and Benchmarks.

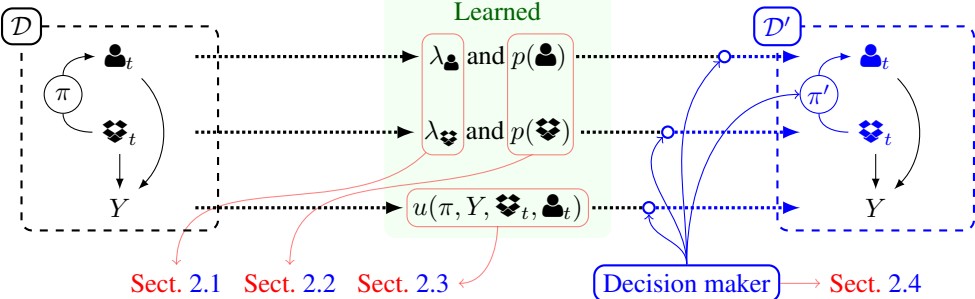

Figure 1: **AllSim overview.** We illustrate how AllSim takes as input a dataset ($\mathcal{D}$) which comprises a set of users (👤) and resources (♻) and outcomes ($Y$). Each of these objects is confounded by an allocation policy $\pi$. From this highly complex dataset, AllSim learns a set of separate components (in green): a distribution for users ($p$(👤)) and resources ($p$(♻)), an associated arrival time based on $t$ ($\lambda_{👤}$ and $\lambda_{♻}$), and a utility ($u$). AllSim then exposes an interface where a decision maker can perturb and influence each component separately, included the allocation policy itself. We highlight these perturbations in blue, resulting in a new dataset ($\mathcal{D}'$) used to measure the effect of each perturbation.

allocations. We note that the above problem scenario is incredibly general. To have an idea of the diversity of situations described as such, we refer to Table 1 where we list a few example situations.

**Resource allocation.** We identify three main challenges one has to overcome when solving problems described by the above. (i) Resources and users are described by multiple (possibly continuous) variables resulting in them being diverse and having complex interactions. (ii) The above are *dynamic non-steady-state scenarios*, which means that at any time the arrival of resources and users may change, the user-specific as well as system-wide utility may change, and even the users and resources themselves may change. (iii) These are multi-user problems, which means that each decision needs to take into account the resource recipient alongside every other user in the system.

**Evaluation.** Given the complex interactions between diverse resources and users, more and more we have to rely on machine learning based allocation policies which model these interactions to optimise a (system-wide) utility [1–4] (cfr. Table 1). While these novel policies receive a lot of research attention, the way in which they are evaluated seems to receive much less while being equally important. In fact, literature introducing these new methods fail to evaluate them against the challenges listed above. We believe the reason is the lack of proper evaluation tools; to our knowledge, there only exist tools that: *Have no diverse resources/users* [5, 6], *Remain steady-state* [7], or *Model single-user systems* [8]. None of them capture the challenges described above.

**Synthetic versus real data.** Another major consideration is the usage of data when evaluating policies. Whenever a real-world dataset is available— comprising users, resources, assignments and outcomes —we have to consider the fact that this dataset is *tainted* by an observational (in-place) policy. We have illustrated this as ♻$_t$–$\pi$→👤$_t$ in Figure 1, where the policy is denoted as $\pi$. The moment we want to test a policy which is different from the observational policy (e.g. $\pi'$ in Figure 1), we deviate from the original dataset as the resource to user assignments are (by definition) different.

A solution could be a completely synthetic simulation to evaluate policies. However, given the detailed and diverse descriptions of resources and users, this would introduce too much bias into our evaluation as every detail needs to be manually specified [9]. The latter is of particular importance when testing these novel ML-based policies, as this is exactly what they were built for in the first place.

**Our solution is AllSim.** Illustrated in Figure 1, AllSim learns separate (unbiased) components from historical data which was biased by a previous observational policy, $\pi$. These components are exposed as an interface which a decision maker can use to modify the system to fit their purpose. An obvious example of such a modification is to replace the past policy with a different policy. Other examples could be changing the arrival rates of users, resources, changing user and resource types, or even resource efficacy by changing the outcomes ($Y$) while still maintaining detail and realism. These perturbations are illustrated in blue in Figure 1. From AllSim, we sample a new dataset to measure the effect of the practitioner's modifications on utilities such as fairness, survival, waste, etc.

Table 1: **Example situations.** We list a few example situations that follow the general problem formalism introduced in this paper and the repeated allocation policy used to solve them.

| Problem setting | Users | Resources | Allocation policy | Utility |
|---|---|---|---|---|
| *Headhunting* | Openings | Applicants | Assignment | Hires (& retention) |
| *Project staffing* | Projects | Workers | Staffing | Project success |
| *Organ transplantation* | Patients | Donors | Matching | Post/Pre TX survival |
| *Mechanical ventilation* | Patients | Ventilators | Triaging | ICU Discharge |
| *Bicycle sharing* | Docks | Bicycles | Redistribution | Idle times |

**Why are such simulators important for ML research?** The successes in other subareas in machine learning are driven largely by the existence of capable and qualitative simulators [10–13]. With an easy-to-use interface and easily customisable environments, simulators allow researchers to focus on model development rather than creating their own (often conflicting) evaluation protocols. With AllSim, we hope to drive innovation for resource allocation in multi-user problems in healthcare, engineering, economics, etc. The aforementioned simulators are great examples of systematic evaluation across entire research communities, however, they do not: learn realistic and unbiased simulation objects from data, allow for multi-user simulation, or model dynamic non-steady-state scenarios.

**Desiderata.** From Figure 1 we identify three important desiderata: (1) A simulation should extract unbiased components from historical data which was tainted by existing policies; (2) The simulation should infer unbiased outcomes despite having access to only these biased data, which includes long-term impact on system-wide utilities since present allocations influence future allocations, requiring counterfactual inference (to determine outcomes under different allocations). (3) Using the extracted components from (1), a user must be able to perturb and change the components to fit their specific needs to evaluate different policies and settings before being deployed in the real world.

**Contributions** In this work, we present AllSim (Allocation Simulator), a general-purpose open-source framework for performing data-driven simulation of scarce resource allocation policies for *pre-deployment* evaluation. We use modular environment mechanisms to capture a range of environment conditions (e.g. varying arrival rates, sudden shocks, etc.), and provide for componentwise parameters to be learned from historical data, as well as allowing users to further configure parameters for stress testing and sensitivity analysis. Potential outcomes are evaluated using unbiased causal effects methods: Upon interaction with a policy, AllSim outputs a batch dataset detailing all of the simulated outcomes, allowing users to draw their own conclusions over the effectiveness of a policy. Compared to existing work, we believe this simulation framework takes a step towards more methodical evaluation of scarce resource allocation policies.

In Appendix B we compare against other strategies used to evaluate allocation policies. AllSim's itself is built using ideas from various fields in machine learning which we also review in Appendix B. Furthermore, in Appendix B.1 we review some medical simulations which *seem* related, but are not.

## 2 AllSim

Let $X \in \mathbb{R}^d$ denote the feature vector of a *user*, and let $\mathcal{X}(t)$ denote the arrival process of users. At each time $t$, let $\mathbf{X}(t) \coloneqq \{X_i\}_{i=0}^{N(t)} \sim \mathcal{X}(t)$ give the arrival set of (time-varying) size $N(t)$. Likewise, let $R \in \mathbb{R}^e$ be the feature vector of a *resource*, and let $\mathcal{R}(t)$ be the arrival process of resources. At each time $t$, let $\mathbf{R}(t) \coloneqq \{R_j\}_{j=0}^{M(t)} \sim \mathcal{R}(t)$ give the arrival set of (time-varying) size $M(t)$.

While we make no assumptions on how users are modelled, we assume that resources are immediately *perishable*—that is, each incoming resource cannot be kept idle, and must be consumed by some user in the same time step. In organ transplantation, for instance, the time between harvesting an organ and transplanting it ("cold ischemia time") must be minimized to prevent degradation [14–16].

Let $Y_+ \in \mathbb{R}$ be the outcome of a *matched* user, drawn from the distribution $\mathcal{Y}(X, R)$ induced by assigning a resource $R$ to a user $X$. At each time $t$, let $\mathbf{Y}_+(t) \coloneqq \{Y_+ \sim \mathcal{Y}(X_R, R) : R \in \mathbf{R}(t)\}$ give the set of outcomes that result from matching each incoming $R \in \mathbf{R}(t)$ with its assigned $X_R$. Likewise, let $Y_- \in \mathbb{R}$ be the outcome of an *un-matched* user, drawn from the distribution $\mathcal{Y}(X, \varnothing)$. At each time $t$, let the set of outcomes for users who are never assigned a resource be given by

$\mathbf{Y}_-(t) := \{Y_- \sim \mathcal{Y}(X, \varnothing) : X \in \mathbf{X}(t), \neg(\exists t' \geq t)(R \in \mathbf{R}(t'), X = X_R)\}$. (Note that we focus on discrete-time settings (e.g. hours or days), and leave continuous time for future work). Then we have:

**Definition 1 (Repeated Resource Allocation)** Denote an *environment* with the tuple $\mathcal{E} := (\mathcal{X}, \mathcal{R}, \mathcal{Y})$. The *repeated resource allocation* problem is to decide which users to assign each incoming resource to—that is, to come up with a *repeated allocation policy* $\pi : \mathbb{R}^e \times \mathcal{P}(\mathbb{R}^d) \to \mathbb{R}^d$, perhaps to optimize some utility defined on the basis of (un-)matched outcomes. For instance, if $Y$ is a patient's post-transplantation survival time, we might wish to maximize the average survival time.

With the necessary notation, and a formal definition of a policy's input and output in Definition 1, we are equipped to introduce each component of AllSim as illustrated in Figure 1. In Sect. 2.4 we also discuss how AllSim's output can be used to evaluate a new policy (or any other modification from the decision maker). Details regarding the simulation life-cycle can be found in Appendix A.

## 2.1 Arrival of users and resources —————————————— $\lambda_{\text{\Large\&}}$ and $\lambda_{\text{\Large\&}}$

There are two necessary ingredients that comprise the arrival of new users, and new resources: the amount ($N(t)$ and $M(t)$, respectively), and the description ($X_i$ and $R_j$, respectively). Each is modelled differently. Before we sample the user and resource description, we first sample the amount of each arriving at time $t$ from an associated arrival process— i.e., in this subsection we will focus on $N(t)$ and $M(t)$. We first introduce the structure of the arrival processes, and explain how their parameters can be learned from data and modified by a decision maker to setup the environment.

First, we stress that $N(t)$ and $M(t)$ are not necessarily sampled from *constant* arrival processes. Instead, we want the user and resource arrivals to change over time either completely or per user/resource type, which we will explain in more detail below. To accommodate this, we split each arrival process into a product of separate arrival processes which we combine into $\mathcal{X}(t)$ and $\mathcal{R}(t)$ as:

$$\hat{\mathcal{X}}(t, \theta_x) = \hat{\mathcal{X}}_1(t, \theta_{1,x}) \times \cdots \times \hat{\mathcal{X}}_K(t, \theta_{K,x}), \tag{1}$$

$$\hat{\mathcal{R}}(t, \theta_r) = \hat{\mathcal{R}}_1(t, \theta_{1,r}) \times \cdots \times \hat{\mathcal{R}}_L(t, \theta_{L,r}), \tag{2}$$

where each individual arrival process in $\hat{\mathcal{X}}_k$ and $\hat{\mathcal{R}}_l$ is parameterised with (learnable) parameters $\theta_{k,x}$ and $\theta_{l,r}$ with $k \in [K]$ and $l \in [L]$, respectively. Each factor corresponds with some (learned or predefined) user-type (Equation (1)) and resource-type (Equation (2)). Having these factors allows us to model increasing numbers of, for example, older/younger patients entering a transplant wait-list.

In order for $\mathcal{X}(t)$ and $\mathcal{R}(t)$ to change over time, we let their parameterisation, $\theta_x$ and $\theta_r$, change in $t$. As an example, we can set the arrival processes to Poisson processes (we refer to Appendix E for other examples) with arrival rates $\theta_x = \lambda_k(t)$ and $\theta_r = \lambda_l(t)$, which we can both model over time as,

$$\lambda_k(t) = \nu_k \lambda_k(0) g_k(t), \tag{3}$$

$$\lambda_l(t) = \nu_l \lambda_l(0) g_l(t), \tag{4}$$

with $\lambda_k(t), \lambda_l(t) \in \mathbb{R}_+$, and $\nu_k, \nu_l \in \mathbb{R}_+$ as a normalising constant such that the sum of all $\lambda_k$ equal some overall arrival rate $a_x$, and similarly, the sum of all $\lambda_l$ equal some overall arrival rate $a_r$. Lastly, $g_k$ and $g_l$ are continuous functions that simulate a user-specified drift. Note that these $g$ can also be a combination of multiple drift scenarios, or can be shared across different $k, l$. Having $g$, allows practitioners to very accurately describe the non-stationarity they wish to test for. Optionally, $\nu_k$ and $\nu_l$ can be kept fixed throughout the simulation such that $a_x$ and $a_r$ vary as does $g_{k,l}(t)$, or it can be recomputed for every step $t$, such that $a_x$ and $a_r$ are kept fixed throughout the simulation.

As such, we have a set of arrival rates, $\Lambda_x = [\lambda_1, ..., \lambda_K]$, with $\sum_k \lambda_k = \alpha_x$ with $\alpha_x \in \mathbb{R}_+$ as the total arrival rate of recipients. The advantage of splitting $\alpha_x$ into multiple $\lambda_k$, is that we can finetune the arrival of certain recipient types, yet allow comparison between $\alpha_x$ and $\alpha_r$ (the total arrival rate for resources). For example, the $k^{\text{th}}$ recipient type may be completely absent when a policy is launched, but over time it gradually enters the system, increasing $\alpha_x$ as a whole. Naturally, we also model the arrival of resources as we have for recipients, but left it out of discussion for clarity.

Learning $\theta_{x,r}$ naturally depends on the choice of arrival process. In our setups below we use a Poisson process and have either: (i) learned the dynamic parameters $\lambda_{k,l}(t)$ as in Equations (3) and (4) using polynomial regression over a time-windowed average of incoming users and resources— over all

data to compute a correct $\nu_{k,l}$, as well as per predefined condition; or (ii) have predefined an arrival function and drift functions, $g$, to illustrate a scenario where one wishes to test a prespecified scenario.

## 2.2 New users and resources ——————————— $p(\text{👤})$ and $p(\text{📦})$

From $\hat{\mathcal{X}}(t)$ and $\hat{\mathcal{R}}(t)$ we sample $N(t)$ and $M(t)$, respectively. Of course, we need to provide the tested policies with more than just an *amount* of users and resources arriving at time $t$. Furthermore, when working with user and resource types (using the decomposition in Equations (1) and (2)), we have $N(t) = \sum_k N_k(t)$ and $M(t) = \sum_l M_l(t)$, where each $N_k(t)$ and $M_l(t)$ represents an amount of users and resources *per type*. As such, we need these types to sample detailed descriptions of each.

When a recipient or a resource arrives, we sample them from a distribution denoted $p_{\theta_x}(X)$ for the recipients, and $p_{\theta_r}(R)$ for the resources. These distributions are either learnt from data, or shared as an open-source (but privatised) distribution. For the user-distributions we learn from $\bigcup_t \mathbf{X}_t$, and similarly, for the resource-distributions we learn from $\bigcup_t \mathbf{R}(t)$. Since both remain independent from the past policy (no policy determines which users and resources arrive in the system), we can use any (conditional) generative model to learn these distributions as we are not required to de-bias these data.

Of course, we need to be able to sample specific user and resource *types*. For this we require *conditional* generative models, where the condition corresponds with a type: $p_{\theta_x}(X)$ becomes $p_{\theta_x}(X|k)$, and similarly $p_{\theta_r}(R)$ becomes $p_{\theta_r}(R|l)$. In case we wish to use an unconditional generative model, we can simply learn multiple: $p_{\theta_{k,x}}(X)$ for each $k \in [K]$, and similarly for $p_{\theta_{l,r}}(R)$ for each $l \in [L]$.

Interestingly, we do not need to account for any variability (learned nor specified) over time, since this is completely modelled through the arrival processes in Sect. 2.1. In particular, whenever we wish one type, $k$ to dominate others, we simply increase $g_k(t)$ in Equations (3) and (4). In case we only want one type to appear after $t$ in the simulation, we set $g_k = 0$ for $t' < t$ and increase it for $t'' > t$.

## 2.3 Utility ——————————— $u(\pi, Y, \text{📦}_t, \text{👤}_t)$

The final component in AllSim, as per Figure 1, are the utilities: functions of the policy ($\pi$), the users and resources ($X$ and $R$), and crucially, the allocation outcomes ($Y$). Given the previous sections, all that remains are the outcomes and how we can combine each element into a new dataset, $\mathcal{D}'$, with *counterfactual* outcomes, $Y'$.

As the outcome is a function of the resource and its recipient, inference is a hard problem as allocations suggested by the tested policy deviate from historical data which was collected under a different policy (i.e., they are counterfactual). Consequentially, some combinations are less observed in the original data, illustrated in Figure 2. In Figure 2 we illustrate two policies, $\pi$ and $\pi'$ which result in different datasets $\mathcal{D}$ and $\mathcal{D}'$. The latter ($\mathcal{D}'$) is what we wish to provide with AllSim, using only data from the former ($\mathcal{D}$).

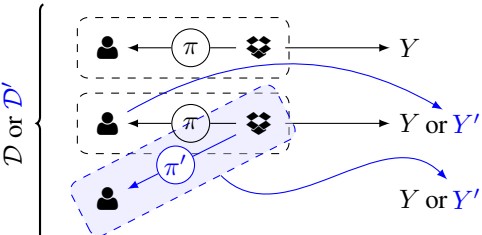

Figure 2: **Allocation policies bias data.** Above illustration depicts two policies: $\pi$ and $\pi'$. Each policy is tasked with assigning resources to users as per Definition 1. Besides users and resources, we observe an outcome, $Y$. Despite observing the same users and resources, a different policy results in *completely* different outcomes, $Y'$, and data, $\mathcal{D}'$.

**Counterfactual inference.** AllSim handles this difficult problem by using a counterfactual estimator. Counterfactual methods correct for allocation bias explicitly [2]. In particular, these methods aim to make an unbiased prediction of *the potential outcome*, associated with some treatment (or resource). We are interested in counterfactual methods that model the potential outcomes for the recipients when they are (not) allocated a resource. A counterfactual estimator then "completes" the dataset ($\mathcal{D}'$) as,

$$\mathbf{Y}(t) = \mathbb{E}[\hat{\mathbf{Y}}(\mathbf{R}(t))|\mathbf{X}_\pi(t)], \tag{5}$$

where $\hat{\mathbf{Y}}(\mathbf{R}(t))$ is the estimated potential outcome, using methodology known in the potential outcomes literature [17–22], and $\mathbf{X}_\pi(t)$ are the recipients selected by a policy $\pi$ at time $t$. The potential outcome is a random variable depicting the (possibly alternative) outcome when the user receives the resource, $R(t)$. Note that this is not the same as simply conditioning the outcome

variables on the users, for the reasons outlined above: conditioning using only biased data will lead to biased estimates for the outcome variable. Hence, literature on counterfactual inference introduced the potential outcomes notation in Equation (5) to differentiate between $Y(R(t))$ and $Y|R(t)$. We provide a comprehensive overview of counterfactual methods and literature in Appendix H.

## 2.4 Putting it all together

We have now discussed each component in the middle section of Figure 1. What remains are the decision maker's perturbations, and finally, combining each component into a new dataset, $\mathcal{D}'$.

**Perturbations.** From Figure 1 we learn that a decision maker can make three types of perturbations: (i) they can replace the original policy, $\pi$, with a new (alternative) policy, $\pi'$; (ii) they can change the utility function, $u$, which takes as argument a dataset comprised of users, resources, outcomes, and a policy; and lastly, (iii) they can change the types, as well as the amount, of users and resources entering the system. Given these perturbations, the policy is allowed to act in a different environment.

Changing the policy in (i) is done simply by implementing the new policy according to the simulation interface (discussed in the next section). We stress once more that this paper does not provide guidance for allocation policies nor does it propose a new policy of any kind. In fact, the policies used in the following section are tried and tested policies, currently in use in practice. Changing the utility function for (ii) is easily done in AllSim as running the simulation does not depend at all on the chosen utility function! As AllSim provides a completely counterfactual dataset, $\mathcal{D}'$, the utility is computed *post-hoc* which allows us to always fall back on the generated dataset. Finally, perturbing arrivals (iii) is already discussed in Sect. 2.1; the arrival processes are perturbed through $g$.

**Sampling data.** Equations (1) to (4) provides us with $\mathbf{X}(t)$ and $\mathbf{R}(t)$. Equation (5) provides us with an estimated potential outcome given $X(t)$, $R(t)$ and their allocations using $\pi'$. AllSim then carefully indicates a timestamp for each combination and then presents the decision maker with a new dataset:

$$\mathcal{D}' \coloneqq \{(X, R, \hat{Y}(R), t)_i : i = 1, \dots, N\}.$$

Having a new (counterfactual) dataset based on $\pi'$, $\mathcal{D}'$, allows to easily calculate various performance utilities of interest, which the decision maker can use to evaluate the allocation policy, *pre-deployment*:

**Definition 2 (Pre-Deployment Evaluation)** Let $f : \prod_k \mathbb{R}^k \times \dots \to \mathbb{M}$ denote an *evaluation metric* mapping a sequence of outcomes $\{\mathbf{Y}(t)\}_{t=1,\dots}$ to some space of *evaluation outcome* $\mathbb{M}$ (e.g. for the average survival time, this would simply be $\mathbb{R}$), where $\mathbf{Y}(t) \coloneqq \mathbf{Y}_+(t) \cup \mathbf{Y}_-(t)$. Given a problem $\mathcal{E}$ and policy $\pi$, the *pre-deployment evaluation* problem is to compute statistics of the distribution $\mathcal{F}_{\mathcal{E},\pi}$ of evaluation outcomes $f(\{\mathbf{Y}(t)\}_{t=1,\dots})$; commonly, this would be the mean $\mathbb{E}_{\mathcal{E},\pi}[f(\{\mathbf{Y}(t)\}_{t=1,\dots})]$.

Note that we have defined $f$ in terms of the *sequence* of per-period outcomes such that it gives maximum flexibility: Depending on how individual outcomes $Y$ are defined, we can measure point estimates (e.g. the mean survival), compare subpopulations (e.g. whether some types of recipients systematically receive more favourable outcomes), examine trends (e.g. whether outcomes degrade as the types of recipients arriving change), or potentially investigate more complex hypotheses.

## 3 AllSim Interface & Examples

Given the formal definition of AllSim presented in Sect. 2 (and further in Appendix A), we now introduce AllSim's programming interface and use it directly to provide some experimental results. We split this section in two major parts: first, we show the type of analysis AllSim can do for us, as well as how to tailor AllSim to the decision maker's needs; and then, we show how realistic the AllSim simulations are, compared to the factual data; we show that AllSim models realistic systems.

### 3.1 Example analysis and decision-maker specifications

As a first example, let us showcase an analysis to illustrate the possible impact AllSim may have in practice. Throughout this section, we will use the open-access United Network for Organ Sharing (UNOS) dataset which comprises 25 years of liver-to-patient allocation. Importantly, we had to make zero adjustments to our framework to fully capture these data, showcasing the generality of the

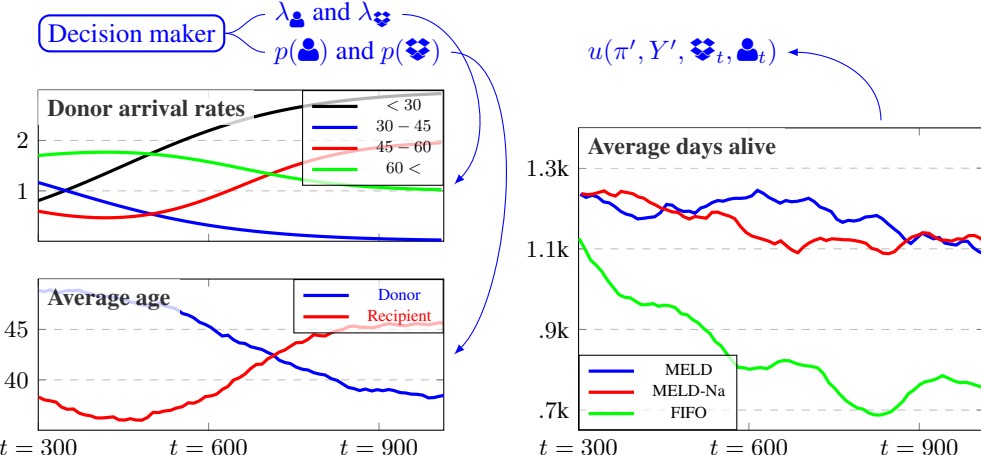

Figure 4: **Specifying a simulation using AllSim.** In the above, a decision maker defines a set of donor arrival rates, based on age ($\lambda_{\bullet}$ and $\lambda_{\heartsuit}$). Using these very simple, but custom, arrival rates, we see a direct influence in the user and resource distributions ($p(\bullet)$ and $p(\heartsuit)$). These perturbations constitute as perturbations of type (iii) as per Sect. 2.4. Finally, the decision maker tries out three different policies: MELD, MELD-na, and FIFO, which constitute as perturbation type (i). The result of these policies is shown on the right. The reported averages are windowed over 300 samples.

AllSim framework. We only use UNOS data until 2019 which, interestingly, predates the COVID-19 global pandemic. As such, it is impossible to evaluate policies using only these data: we need AllSim to model a counterfactual scenario that mimics what we saw during the pandemic to test a policy.

In Figure 3 we ran the MELD-Na policy in two hypothetical scenarios: one where COVID-19 happens (which resulted in a 50% drop in the donor liver arrival rates [23–25]), and one where it doesn't. With AllSim we can model each scenario confidently. For this particular example, we fix the seed of AllSim and only change the supply of organs by giving two different resource arrival processes:

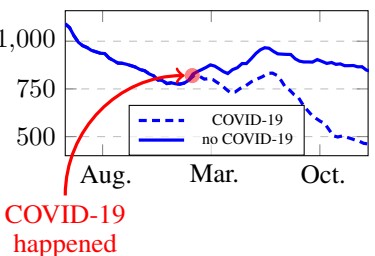

**Two hypothetical scenarios**

Figure 3: **Two hypothetical scenarios.** We require AllSim to evaluate a policy (e.g. MELD-Na) in hypothetical (counterfactual) scenarios. The x-axis is time, and the y-axis indicates survival time.

```
1   def covid(t):
2       if t < 600:
3           return .5
4       else:
5           return .25
6
7   def no_covid(t):
8       return .5
```

Having illustrated the power of AllSim, let us now show how a decision maker may introduce their perturbations (such as the `covid` and `no_covid` arrival processes from above) into the AllSim simulator. For this, we will provide the donor-organ system with two specific perturbations: (1) we will change the policies (from MELD, to MELD-na, and a simple FIFO policy), and (2) we will increase the user age and decrease the donor age. These two perturbations respectively illustrate perturbation types (i) and (iii), listed in Sect. 2.4 (recall that perturbation type (ii) was changing the utility which is done *after* a counterfactual dataset is sampled and hence does not require testing).

Consider Figure 4 which shows the resulting simulation and the found utility when perturbing the organ arrival rates as well as the allocation policies. The take-away from this experiment is not the performance of the policy (although, reassuringly, MELD and MELD-na[1] do outperform FIFO). Instead, we learn that increasing user age results in dropping the survival time, regardless of the donor age which is decreasing. This is not surprising, as older transplant patients simply have less time to live, whether they get an organ or not. Which leads to the next question: is AllSim realistic?

---

[1]Which are the policies used in Europe and the USA for donor liver allocation.

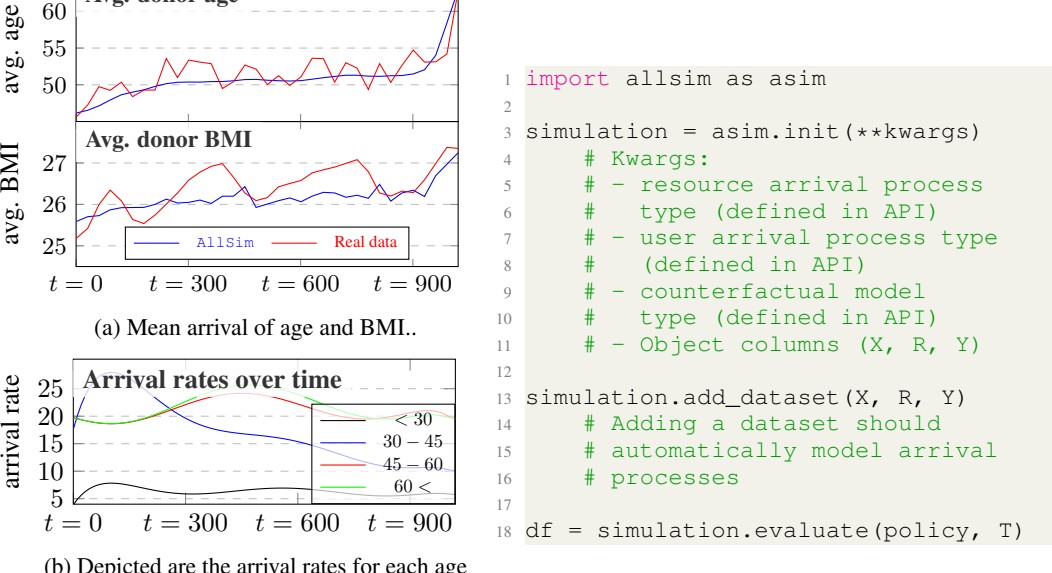

```
1   import allsim as asim
2
3   simulation = asim.init(**kwargs)
4       # Kwargs:
5       # - resource arrival process
6       #   type (defined in API)
7       # - user arrival process type
8       #   (defined in API)
9       # - counterfactual model
10      #   type (defined in API)
11      # - Object columns (X, R, Y)
12
13  simulation.add_dataset(X, R, Y)
14      # Adding a dataset should
15      # automatically model arrival
16      # processes
17
18  df = simulation.evaluate(policy, T)
```

(a) Mean arrival of age and BMI..

(b) Depicted are the arrival rates for each age bracket. Note that, in this instance, we learnt these from data, but one can use *any* function.

Figure 5: **AllSim (easily) simulates realistic environments.** Using real-world data on donor organs, we let AllSim model 3 years of organ arrivals and compare it with the actual arrival as reported in the data. In Figures 5a and 5b we show AllSim's output (in donor age and BMI), given the code on the right. With minimal code, a simple condition (4 age brackets), and conservative models (polynomial regression to fit the arrival rates, and a Gaussian kernel density to model the organ densities), we find that AllSim accurately models the actual (real-world) arrival of organs as reported in the UNOS data.

## 3.2 AllSim's realism

In this section, we will learn an AllSim configuration purely from the UNOS data (i.e. without a decision maker's input), such that we can compare AllSim's output side-by-side with what actually happened in UNOS. If they match up, we confirm that AllSim can output realistic scenarios (as UNOS is a real dataset). However, before we do so, we first show *how* we use AllSim from a programming perspective and configure appropriate arrival processes and densities for this particular use case.

First we determine how many users and resources we need to sample, once we know the amount we sample them from a generative model. The former is modeled through a Poisson arrival rate that changes over time, and the second is sampled from some learnt density. Of course, a user can implement their own arrival process by inheriting from the abstract `ArrivalProcess` class.

Importantly, we need to be able to condition the density on some pre-specified characteristic of the object of interest. For example, one may be interested in modelling the arrival of harvested organs of older patients distinctly from younger patients. An example of this is provided in Figure 5, where we show the changing resources coming in the system, alongside the code that generated the result.

**Object densities.** Before discussing a temporal arrival rate, we first discuss modelling the object's densities. Consider `lns 3-12` in the righthand side of Figure 5. Using this code, we first define what we want to condition on, using a `Condition` object: in this case we formulate age brackets. With the `KDEDensity` class, which is a subclass of the abstract `Density` class, we can automatically model a density, conditioned on these age brackets. Each `Density` object implements a `fit` and `sample` function, which is used to sample new objects by the `System`, which we discuss next.

**Arrival processes.** Using a `Density`, we move on to `lns 14-26`, where we first build a system of multiple arrival processes, one for each discrete condition as in Equations (1) and (2). In particular, we define a `PoisonProcess` for each condition (or age bracket), which is then provided to a `PoissonSystem`. Using the `PoissonSystem`, we can sample the arriving objects for each $t$ in

ln 25. Note that we also model `alpha`, returning the overall arrival rate, such that the system can calculate an appropriate $\nu$. Figure 5a shows that `AllSim` accurately models the arriving objects.

With the arrival processes coded above together with a counterfactual `Inference` object, we compose a `Simulation` object— the main interaction interface. In particular, one defines a set of arrival rates (such as in Figure 5b) for both users and resources (cfr. `ln 13-17`) to create a simulation:

```
1 simulation = asim.init(resource_process, patient_process, inference,
      columns)
```

With that simulation, a practitioner can instantiate a `Policy`, which implements the `add` and `select` methods. For example, we have implemented the MELD policy [26], which is a widely known and used ScRAP for liver allocation. Using the `simulation`, we can generate a simulated dataset:

```
1 df = simulation.evaluate(policy=meld_policy, T=T)
```

Where `df` is a Pandas `DataFrame` [27, 28]. Naturally, `df` contains an enormous amount of information w.r.t. the `policy`'s allocations in our environment. As such, we have included only a subset of the potential results in Figure 4. Additional results and details can be found in Appendices C and H. Ultimately, the practitioner determines appropriate analysis, settings, and performance metrics.

### 3.3 Beyond Organs

AllSim is a general purpose simulator which evaluates scarce resource allocation policies. While we have mainly focused on organ-transplantation so far, AllSim is also applicable in other settings. To illustrate, we show how one can implement a vaccine distribution policy evaluation system in AllSim. This use-case illustrates the few adjustments one has to make compared to the organ-allocation problem. Specifically, in vaccine distribution, each resource is the same and arrives in batches. Furthermore, the type of patient-in-need is also much broader (in fact, they cover the entire population). Yet, AllSim is perfectly capable of modelling this scenario given the following:

- Batch arrival requires a multiplier: if the Poisson process samples a value of 2 on one day, we could simply interpret this as two batches of 1000 doses, i.e. multiply by batch content.

- We no longer require a resource density as vaccines are not unique, contrasting organ allocation. This is implemented as a dummy-density that always returns 1 (or the vaccine amount).

- The broader patient-type is achieved by retraining the recipient-density on the entire population.

These implementation details are relatively simple to implement using AllSim's modular API. While not necessarily a problem in vaccine distribution, recipient arrival in the ICU in a setting of infectious disease (such as COVID-19), is definitely different compared to the organ-allocation setting. With organ-allocation, we can safely assume a Poisson process for recipient arrival as recipients enter the system independently. This is not true for infectious diseases: one recipient arriving may indicate higher infection rates. As such, recipients *do not* arrive independently, motivating AllSim's design.

It is clear that above scenario can no longer rely on a Poisson arrival process for new recipients entering the system. Instead, accurately modelling a situation of infectious disease could be done using a Hawkes process. To illustrate, we include some code below showing exactly how one may go about including such a Hawkes process in AllSim (replacing the Poisson processes used earlier).

```
1    class HawkesProcess(PoissonProcess):
2        def __init__(self,
3                lam: float=.1,
4                update_lam: Callable[[int], float]=lambda t: t,
5                delta: float=.1,
6                a: float=.2):
7            assert a >= 0, "a should be larger than or equal to 0"
8            assert delta > 0, "delta should be larger than 0"
9
10           super().__init__(lam, update_lam)
11
12           self.a, self.delta, self._samples = a, delta, []
13
```

```
14        def get_lam_unnormalized(self, t: int) -> float:
15            return self._baseline_lam + np.sum(
16                self.a * self.beta * np.exp(-beta * (t - self._samples[
    self._samples < t])))
17
18        def progress(self, t: int, neu: float=1) -> int:
19            self.lam = neu * self.get_lam_unnormalized(t)  # eqs. (5, 6)
20            sample = np.random.poisson(lam=self.lam)
21            self._samples.append(sample)
22            return sample
```

**Allocation policies from machine learning and OR.** It seems that both the ML [1–3, 29–34] and OR [35–41] community is focused more and more on this important class of problems– which is fantastic! But it also warrants careful evaluation. Furthermore, if we find that the evaluation strategies in medicine (which generally propose linear combinations of features [26, 42] or simple CoxPH models [43–46]) have shortcomings, then this is certainly the case for much more complicated strategies introduced in ML or OR. In fact, a recent survey confirmed exactly this concern: [47, cfr. Limitations of ML in transplant medicine]. It is in these extended scenarios where AllSim could help.

Naturally, problems solved by the OR community concern a *variant* of the general problem presented in this paper. For example, Balseiro et al. [40] are concerned with distributing a *fixed* set of resources, to a varying set of incoming users. While different, such problems can still be modelled in AllSim. In the specific case of Balseiro et al. [40], resources are not unique (they represent an amount) and require much less machinery than what we require to model the varying resource scenario. Specifically, one can model the remaining amount of resources as an attribute in our `Policy` class.

## 4  Conclusion

AllSim provides the means to perform standardised evaluation of repeated resource allocation policies in non-steady-state environments. While our experiments focus on organ-transplantation for the sake of exhibition, Appendix G illustrates AllSim for COVID-19 vaccine distribution, an example outside organ-transplantation. We believe that AllSim's generality and modularity allows for *sensible* adoption in a wide range of application areas. Furthermore, having standardised evaluation will encourage research in this very important and impactful domain spanning many application areas.

Conducting research in repeated resource allocation requires consideration of a policy's societal impact. While we believe AllSim will *aid* (rather than negatively impact) in this respect (by offering more than simple aggregate statistics), in Appendix F we provide a section dedicated to this topic.

**Ethical research.** We envisage AllSim as a tool to *help* accurate and standardised evaluation of repeated resource allocation policies, however emphasise that any finding would need to be further verified by a human expert or in some cases by a clinical trial. Ultimately, the decision on whether or not to trust a decision making tool is up to the acting decision-maker and ethics board. We hope that AllSim can help in any way to facilitate that decision, but stress that suggestions or evaluation always require critical assessment, as is the case for any research. We also refer the reader to Appendix F for a more thorough discussion on the potential societal impact of systems such as AllSim.

**Reproducibility.** To encourage reproducibility, we have included all our code to reproduce the presented results (as well as those in Appendix C). It should be clear from this paper, that reproducibility is actually one of the main reasons for doing this type of research in the first place. Furthermore, we have included a detailed discussion on how to use our simulation in Appendices D and E.

## Acknowledgements

We would like to thank our many collaborating clinicians, and in particular, Alexander Gimson, for many interesting discussions leading to this work. JB is funded by the W.D. Armstrong Trust, DJ is funded by Alzheimer's Research UK (ARUK), and AC is funded by Microsoft Research.

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
