# Appendix: AllSim

## Table of Contents

## A  Simulation Life-cycle

Using AllSim's definition in Sect. 2, we will formalise a high-level simulation life-cycle. In simple terms, the simulation loop will model the interaction between $\mathcal{E}$ and the `policy` for each consecutive day ($t = 1, 2, \dots$) until some pre-specified end ($T$). We have organised this section around Algorithm 1 where we connect each important line with a component in Figure 6 and discuss it in a separate subsection. In Figure 6 we illustrate the interaction between each component, and clarify which parts of the simulation can be learned, and which parts should be parameterised by the practitioner.

### A.1  (i) `Sample users and resources`

The first step in Algorithm 1 comprises the first and second component in Figure 6, repeated below,

$$\underbrace{\lambda_x, \lambda_r \overset{\text{\tiny \faUsers}}{\leadsto} t'}_{\text{arrival amount}} \underbrace{\left(\#\right) \overset{\sim}{\leadsto} \mathbf{X}(t), \mathbf{R}(t)}_{\text{sample objects}}.$$

Essentially, the goal in `ln 4` of Algorithm 1 is to translate the arrival rates to a set of users and resources, trademarked by their features sets. With above formulation, we model the arrival processes as a two-step approach: (1) sample the amount of objects we may expect, (2) sample the objects from a distribution. Indicated in Figure 6, only some of these component are learned, and others user-defined. While not limited to, our examples focus on tabular data. Hence, any density strategy

[Figure]

Figure 6: **Simulation life-cycle of AllSim.** In the leftmost part of this figure, we have illustrated the simulation life-cycle. In the rightmost part, we list each component and indicate whether or not they are learned from data. *Simulation life-cycle.* First, we sample the amount of each object arrives on day $t$. The expected amount is indicated by $\lambda_x$ and $\lambda_r$ for the users, and resources, respectively. The expected amount should be interpreted as an arrival rate. Once the amount of objects is specified, they are sampled from a distribution $\mathbf{X}(t)$ and $\mathbf{R}(t)$. Next, each objected is presented to the policy, we wish to test, and the policy replies by selecting a set of users whom the resources will be presented to. Finally, the simulation evaluates the policy's decision, by providing an inferred outcome. Wrench icons (🔧) and an ✗'s in the table indicate which components can be user-specified, others (with ✓) are learned. Finally, $\mathbf{X}(t)$, $\mathbf{R}(t)$, and $\mathbf{Y}(t)$ compose a dataset $\mathcal{D}'$ for evaluation.

---

**Algorithm 1:** *Main simulation loop.* This simulation life-cycle acts as a section overview for Appendix A. In our main text we discuss how each line is simulated.

**input** :Environment, $\mathcal{E}$; A resource allocation policy denoted as `pol`
**output** :Policy runtime summary
1 Start, $t = 0$;
2 **while** *simulation runs* **do**
3 $\quad$ $t \leftarrow t + 1$; $\qquad\qquad\qquad\qquad\qquad\qquad\qquad$ /* **iterate time** */
4 $\quad$ $\mathbf{X}(t), \mathbf{R}(t) \sim \mathcal{E}(t)$; /* **(i)** sample users and resources (Appendix A.1) */
5 $\quad$ `pol.add`$(\mathbf{X}(t))$; $\qquad$ /* **(ii)** notify policy of users (Appendix A.2) */
6 $\quad$ $\mathbf{X}_{\mathbf{R}(t)} \leftarrow$ `pol.select`$(\mathbf{R}(t))$;
$\quad\quad$ /* **(iii)** allocate resources (Appendix A.3) */
7 $\quad$ $\mathbf{Y}_t \sim \mathcal{Y}_{\mathcal{E}}(\mathbf{X}_{\mathbf{R}(t)}, \mathbf{R}(t))$; $\qquad$ /* **(iv)** consume resources (Appendix A.4) */
8 **end**

---

should handle such data. This is not an easy challenge, since the structure in tabular data is best respected [9, 48].

**A.2** **(ii)** `notify policy of users`

After the simulation has sampled $\mathbf{X}(t)$ and $\mathbf{R}(t)$, the `policy.add`$(\mathbf{X}(t))$ function is called. Specifically, this notifies the policy of the arrival of new users, which in Figure 6 corresponds to,

$$\mathbf{X}(t), \mathbf{R}(t) \longrightarrow \pi \cdot$$

Note that the above is simply a service provided by the simulation to the policy, and is thus not a learned. The reason for this is, that the organisation of in-need-users is entirely up to the policy. For example, more traditional policies may maintain only one priority queue [44], whereas more novel policies may maintain multiple, based on recipient and resource types [1]. As such AllSim remains agnostic to the tested policy, resulting in a more general-purpose framework.

**A.3** **(iii)** `allocate resources`

Like Appendix A.2, step (iii) is entirely managed by the tested policy, and in Figure 6 corresponds with,

$$\pi \xrightarrow{\text{select}} .$$

Selecting which users that get to consume the resource, is generally what differentiates policies. When a set of users is selected (through a `policy.select`$(\mathbf{R}(t))$ call in the python interface, AllSim retains every piece of information associated with the select-call, which is used to evaluate.

Having the simulation retain all this information, allows AllSim to output a dataset of past policy behaviour, much like the original (real-world) dataset we provided to the simulation in order to learn

Table 2: **Overview of related work.** We categorise our related work in four categories: off-policy learning, causal inference, clinical simulations, simulations for evaluating reinforcement learning agents. For each category we provide the most prominent method of calculating policy performance, and whether the categories take into account four major questions: (i) is the method tunable to new settings?; (ii) can we evaluate using data?; (iii) are the performance estimates unbiased?; and (iv) can we evaluate beyond simple aggregate descriptive statistics?

| | Citations | Perf. calculation | (i) | (ii) | (iii) | (iv) |
|---|---|---|---|---|---|---|
| Off-policy learning | [49–55] | $\frac{p_{\pi'}(R\|X)}{p_\pi(R\|X)}Y \sim \mathcal{D}^\pi$ | ✗ | ✓ | ✓ | ✗ |
| Causal inference | [56–59] | $p(R\|X)^{-1}Y \sim \mathcal{D}^\pi$ | ✗ | ✓ | ✓ | ✗ |
| Clinical simulations | [44, 60–64] | $\mathbb{E}_{\mathcal{D}^\pi}[Y\|X]$ | ✗ | ✓ | ✗ | ✓ |
| Reinforcement learning | [10–13, 65–68] | $\mathbb{E}_{\text{sim}}[v(Y)\|X, R]$ | ✓ | ✗ | ✓ | ✓ |
| AllSim | (ours) | $\mathbb{E}_{\mathcal{D}^{\pi'} \sim \mathcal{X}' \times \mathcal{R}'}[Y\|X]$ | ✓ | ✓ | ✓ | ✓ |

the various components. In essence, with AllSim we are able to sample a synthetic dataset of a counterfactual scenario such that we are able to test a policy of interest, *as if it were already in use*.

### A.4 (iv) consume resources

The final component in AllSim is the inference-component, yielding an outcome after a resource was consumed by a user. Importantly, this component allows us to evaluate a policy, despite it deviating from the data used to learn the simulation. In Figure 6, this component is illustrated as,

$$\mathcal{Y} \rightarrow \mathbf{Y}(t) \leftarrow \pi.$$

The outcome is a function of the resource and its recipient, making inference hard as some combinations are less observed in the original data. Essentially, the tested policy may make out-of-distribution combinations, as the simulation is learnt from data that was collected under some other policy, illustrated in Figure 2 where two policies, $\pi$ and $\pi'$ result in different datasets $\mathcal{D}$ and $\mathcal{D}'$.

**Counterfactual inference.** AllSim handles this by using a counterfactual estimator to infer allocation outcomes as they deal with this issue explicitly. Counterfactual methods aim to make an unbiased prediction of the potential outcome, associated with some treatment (or resource). We are interested in counterfactual methods that model the potential outcomes for the recipients when they are/are not allocated a resource. A counterfactual estimator then "completes" the simulated dataset as,

$$\mathbf{Y}(t) = \mathbb{E}[\hat{\mathbf{Y}}(\mathbf{R}(t))|\mathbf{X}_\pi(t)], \tag{6}$$

where $\hat{\mathbf{Y}}(\mathbf{R}(t))$ is the estimated potential outcome, using methodology known in the potential outcomes literature, and $\mathbf{X}_\pi(t)$ are the recipients selected by $\pi$ at time $t$. Equation (6) provides the recipient-resource pair with an estimated outcome, and presents the practitioner with a dataset:

$$\mathcal{D}^\pi := \{(X, R, \hat{Y}(R), t)_i : i = 1, \dots, N\}.$$

$\mathcal{D}^\pi$ allows to easily calculate clinical measures of performance, as we demonstrate in Sect. 3. More information regarding these counterfactual models is provided in Appendix H.

## B Extended Related Work

We have summarised work related to ours in four major categories: (a) Off-policy learning and evaluation, (b) causal inference, (c) clinical simulations and trials, and (d) simulations for evaluating reinforcement learning (RL) agents. A high-level summary of these areas can be found in Table 2. For a discussion on related work in the clinical domain, we refer to our Appendix B.

**(a) Off-policy evaluation and (b) causal inference.** Data are collected under some active resource allocation policy, which determines how resources are paired with users, which in turn determine the outcomes we get to observe. Clearly, our setting is connected to off-policy evaluation, as we wish to evaluate a policy that is different from the policy that collected the available data. Figure 2 illustrates different matches (depicted as 👤←❤) made across different policies resulting in different outcomes: $Y$ or $Y'$, and datasets, $\mathcal{D}$ and $\mathcal{D}'$. The difficulty of this situation, is that we observe either $\mathcal{D}$ or $\mathcal{D}'$, *not both*, relating to the potential outcomes setup [17, 18, 69]. Using $\mathcal{D}$ to evaluate $\pi$ simply means calculating some descriptive statistics. In contrast, evaluating $\pi'$ on $\mathcal{D}$ is much more involved, as the same statistics would yield a biased estimate.

One way to calculate, for example the average $Y' \sim \mathcal{D}$, is to weigh each sample: $g(\text{👤}, \text{💠})Y \sim \mathcal{D}$. In the literature on off-policy evaluation (and learning), this is known as *importance sampling*, where each sample is weighted according to the likelihood of it belonging in $\mathcal{D}'$. Interestingly, the exact same strategy is also widely known in counterfactual learning and causal inference, under the name *inverse propensity weighting* (IPW) [51]. The key difference between both, is the target distribution. Where importance sampling transforms the estimate from one policy's distribution, to another; IPW transforms from a policy's distribution, to an unbiased estimate (i.e. to a $\frac{1}{N}$ weighting for averages).

Shown in Table 2, neither IPW nor importance sampling, provide a solution to evaluating scarce resource allocation policies. One reason for this is that the outcome is a (weighted) estimate of only one aggregate descriptive statistic (for example, expected outcome or reward). When we want additional estimates, one requires a different weighting scheme [54]. Futhermore, when we wish to test a policy in a different environment, where for example the recipients' or resources' distributions change, evaluating a policy becomes increasingly more difficult if one wishes to rely on IPW or importance sampling due to the increased differences in likelihood density. Furthermore, evaluation goes much beyond aggregate statistics, as we may be interested in, for example, demographic differences between recipients and non-recipients (Appendix C includes an example using AllSim).

**(c) Clinical evaluation and trials.** A naive method to evaluating these policies, is to model $Y'$ using simple (linear [44]) regression. Any combination $\text{👤} \xleftarrow{\pi'} \text{💠}$, through a new policy, $\pi'$ has an estimated outcome $\hat{Y}'$ using a regression model, trained on $\mathcal{D}$. Herein lies the problem: the regression model is still biased to $\pi$. In particular, $\mathcal{D}$ simply does not contain pairs that $\pi'$ would make, and therefor has trouble estimating $Y'$. By using causal methodology (Appendix A.4), this is one key area where AllSim improves upon contemporary clinical simulations [44, 70–72], as these only "replay" the past without the possibility of changing the environment characteristics nor allow counterfactual inference.

Clinicians do have another way to account for this: clinical trials [60, 61]. Clinical trials estimate a causal estimand, which could then be used to "complete" the dataset as a naive regression model would above. However, two problems arise: setting up a clinical trial can sometimes be considered unethical (especially when dealing with scarce resources, and more specific research questions) [73], or clinical trials may suffer from compliance issues which will still result in biased outcomes [74, 75].

**(d) Simulations for evaluating RL agents.** A final set of solutions is that of simulations in which an RL agent can act and learn. Naturally, there are differences between policies for RL and scarce resource allocation, such as the importance of future reward which implies correlated consecutive states, and a fixed action space. However, there are also similarities, such as the definition of a policy (being a set of rules which take into account a context or state), and the online nature of the problem.

Literature on RL is blessed with some of the most well-known and actively maintained libraries for evaluating a wide range of RL algorithms [10, 11, 67]. However, other than being inapplicable in our setting (for the reasons included above), they also have one other major downside: *they do not learn from data*. While these simulations model some very interesting environments, each aspect of the simulation is hard-coded. In order to evaluate a policy that is to be deployed in a real-world setting, we have to test it to the specifics of the environment of interest; to be defined by the practitioners developing the policy. In fact, we consider this a major reason for developing and using AllSim.

AllSim marks a significant advance over the current state-of-the-art: currently *no evaluation technique* allows a practitioner to *change the behaviour of the evaluation environment*, despite the fact that the actual environment most certainly will change in the future; nor do contemporary evaluation techniques account for potential bias from previous policies, resulting in poor performance estimates.

## B.1 Medical simulation

Let us discuss how the literature which introduces novel allocation policies, in particular to donor-organ allocation. From this, we observe that they all come up with their own unique simulation in order to validate their proposal.

**Allocation polices from medicine.** As an example, consider the following papers introduced in the medical domain (we focus here on the liver allocation setting as this is also where we focused on in our paper): [43–46, 76, 77]. Each paper, all coming from different research groups, provide a custom simulation to validate their allocation policy. While some focus only on gathering test data (such as

the recent [46]), others construct a simulation by simply iterating over the patients in the sequence they arrived in reality (such as [44]). While we would note some flaws in the way these policies are evaluated (e.g. counterfactual trajectories when allocations misalign), the truly striking observation is that *each evaluation strategy is different!* Not one paper reuses the same simulation; AllSim may change this going forward.

**Allocation policies from machine learning and OR.** The same is true for the ML [1–3, 29–33] and OR [35–40] communities. It seems that both the ML and OR community is focused more and more on this important problem– which is fantastic! But it also warrants careful evaluation. Furthermore, if we find that the evaluation strategies in medicine (which generally propose linear combinations of features [26, 42] or simple CoxPH models [43–46]) have shortcomings, then this is certainly the case for much more complicated strategies introduced in ML or OR. In fact, a recent survey confirmed exactly that: [47, cfr. Limitations of ML in transplant medicine].

Naturally, there problems solved by the OR community concern a *variant* of the general problem presented in this paper. For example, [40] are concerned with distributing a *fixed* set of resources, to a varying set of incoming users. While different, such problems can still be modelled in AllSim. In the specific case of [40], the resources are not unique (they are represented by an amount) and require much less machinery than what we require to model the varying resource scenario. Specifically, one can model the remaining amount of resources as an attribute in our `Policy` class.

**SimUnet.** While indeed related, SimUnet is in fact very different from AllSim. In particular, from the online README document (see `https://unos.org/wp-content/uploads/1-page-SimUnet.pdf`) it can be seen that SimUnet is more concerned to simulate *offers* from the transplant clinician's point of view; not *allocations* from an overarching healthcare system (such as UNOS, or the NHS).

## C Extended results

### C.1 Running a simulation

**Tuning the simulation.** Running and composing a simulation, is as simple as defining how we wish to let the supply of resources and recipients evolve in $t$. Such evolution is expressed a changing arrival rate (crf. Equations (3) and (4). In particular, one only needs to define $\lambda_{x,r}$, normalisation is handled by `AllSim`. However, even there does `AllSim` offer detailed specification possibilities. Like $\lambda_{x,r}$, we can specify a custom function for $\alpha_{x,r}$ also. Normalisation, will then respect the value of $\alpha$ at time $t$. For example, if we wish the total arrival rate of all resources to remain constant at 5, we simply set $\alpha_r$ to a constant function:

```
1 alpha_r = lambda t: 5
```

This way, no matter how complicated our functions for $\lambda_r$, are, we know that the total arrival rate, across all conditioned distributions for the resources, we remain fixed at 5.

**Inference.** One additional component to `AllSim`, is its `Inference` module. In our simulations, we relied on OrganITE [2], where we adopted the original source-code provided by the authors, to our `Inference` module. Having a causal model, allows to have unbiased estimates for the resource to recipient pairing's outcome (column "Y" in Table 3).

Building a custom `Inference` object, is as simple as inheriting from `Inference`, which requires the user to implement the `infer(x: np.ndarray, r: np.ndarray) -> Any` function.

### C.2 Analysing `AllSim`'s output.

Running `df = simulation.simulate(policy, T=T)` yields a `DataFrame` object, containing each match made by the `policy`, in the environment simulated by the `ss.Simulation` object. Consider Table 3 for an excerpt of the output for which we provided some aggregate results in Figure 4. Table 3 is exactly the type of data one may expect when learning about a policy, or even learning a new (ML-driven) policy [2].

Given the output of our simulation, we can see each variable, both donor as well as recipient progress over time. For example, one may be interested how each gender evolves w.r.t. allocations, with the

Table 3: **Example output.** When donor covariates are `NaN`, we know a patient died on the wait list, as they did not receive a donor organ. Note that, we only display some covariates, we have included code in our submission, should the interested reader want to inspect the complete `DataFrame`.

| Donor covariates | | Recipient covariates | | | | | |
|---|---|---|---|---|---|---|---|
| Age | BMI | INR | Sodium | Creatinine | Gender | Y | Day |
| 28.1628 | 27.7080 | 4.8144 | 138.10 | 11.351 | M | 1792.9 | 1 |
| NaN | NaN | 0.5663 | 137.71 | 0.4535 | F | 5.0 | 7 |
| NaN | NaN | 6.694 | 135.63 | 1.7323 | F | 24.948 | 26 |
| 34.893 | 21.852 | 3.8375 | 134.15 | 7.9500 | M | 257.79 | 1 |
| | | | $\cdots$ | | | | |

[Figure]

Figure 7: **Recipient-gender in function of time.** Given the MELD policy, and the changing supply of resources (cfr. Figure 4, we find that the recipient-gender changes slightly. Note that, this type of analysis is easily done given `AllSim`, as we can easily run analysis on the provided `DataFrame` object, returned by our simulation.

changing supply of resources (as in Figure 4). Naturally, this is but one example of what one can achieve in terms of analysis on MELD (or any other policy), using `AllSim`. As such, we encourage the reader to browse through the provided code, in order to get a sense of what the possibilities really are when using `AllSim`.

# D  Details on `AllSim`

We provide some additional detail on `AllSim`'s functionality, structure, and future goals.

Please find the code for AllSim online at:

https://github.com/jeroenbe/allsim

or

https://github.com/vanderschaarlab

## D.1  Neat functionality

**One-hot encoded variables.** In heterogeneous data, one may expect a mixture of continuous and categorical variables. Good practice to handle categorical variables, is to one-hot encode them into columns of ones and zeroes, for each category. Most density estimation methods, do not automatically sample these mixtures of continuous and categorical data. Despite its performance and simplicity, a standard Kernel Density estimator, like the one we use in our simulation, cannot handle these data out-of-the-box. However, our implementation *does*. One only needs to provide the specific groups of columns which compose the categorical variables.

For example, the gender column for the recipient may be one-hot encoded into: `"GENDER_0"` and `"GENDER_1"`. When fitting our `KDEDensity`, we simply provide these columns to the `one_hot_encoded=[["GENDER_0", "GENDER_1"]]` parameter, and the `KDEDensity` automatically transforms these columns, in such a way that only one of them (being the one with maximum probability), is translated into `1`.

**Auto-scaling.**    Dealing  with  a  normalised  arrival  rate  can  be  tricky  to  detirmine up  front.    As  such,  `AllSim`  does  this  automatically.    By  solely  providing  an

`alpha: Callable[[int], float],`AllSim's `PoissonSystem` objects will normalise any output that the `System`'s `PoissonProcess`es may have. Naturally, normalisation will depend on the chosen process, and as such does not propagate up to the general `System` nor `Process`. If, for example, on wishes to implement an alternative point process (not a Poisson process), normalisation may look differently.

**Automatic conditioning.** Another nice functionality provided in `AllSim` is automatic conditioning, in case no `Condition` is provided. A `Density` does this, by first clustering the data (from which it will later learn its density), into `K` clusters. Naturally, one will have to provide the amount of clusters before the `Density` is able to automatically condition. Having these separate clusters, still allows defining arrival rates and processes for each cluster center, i.e. `K` must match the amount of `Process`es are provided to the `System`.

## D.2 Package structure

The `AllSim` package structure us as follows:

```
1  |_ src
2      |_ infer.py
3      |_ sim.py
4      |_ outcome
5          |_ counterfactual_inference.py
6          |_ counterfactual_models.py
7      |_ policies
8          |_ base.py
9          |_ policy.py
```

The two main components are in `infer.py` and `sim.py`, where `infer.py` contains everything related to `Density`, `Process`, and their subclasses; and `sim.py` concerns the `Simulation` class.

In `outcome` and `policy`, we provide basic implementations of some well known policies and counterfactual models. Note that these are actually non-essential to `AllSim`, as they can be completely user-defined, as long as they respect the `Inference` and `Policy` abstract classes, provided in the respective folders.

**Inheritance.** The base classes (`Density`, `Policy`, and `Inference`) can all be implemented by the user. Consider following class structures:

```python
1      class Density:
2          def __init__(self
3              condition: Condition, # see below for more details
4              K: int=1,
5              drop: np.ndarray=np.ndarray(['condition'])):
6              ...
7
8          @abstractmethod
9          def sample(self, n: int=1) -> np.ndarray:
10             ...
11
12         @abstractmethod
13         def fit(self, D: pd.DataFrame) -> None:
14             ...
15
16     class Policy:
17         def __init__(self,
18             self,
19             name: str,
20             initial_waitlist: np.ndarray,
21             dm: OrganDataModule): # a standardised datamodule should the
    policy want to learn from data
22             ...
23
24         @abstractmethod:
25         def select(self, resources: np.ndarray) -> Tuple[np.ndarray, np.
    ndarray]:
```

```
26              # returns recipients and resources, matched
27              ...
28
29          @abstractmethod
30          def add(self, x: np.ndarray) -> None:
31              # allows policy to add recipients to internal waitlist (if
        needed)
32              ...
33
34          @abstractmethod
35          def remove(self, x: np.ndarray) -> None:
36              # allows policy to remove recipients, when for example they
        die
37              ...
38
39      class Inference
40          def __init__(self,
41              model: Any,      # the Inference class acts as a wrapper for
        any model type
42              mean: float=0,   # mean and std are necessary to scale the
        outcomes, assuming
43              std: float=1):   #   standard scaling
44              ...
45
46          def __call__(self, x: np.ndarray, r: np.ndarray, *args: Any, **
        kwargs: Any) ->Any:
47              return self.infer(x, r, *args, **kwargs)
48
49          @abstractmethod
50          def infer(self, x: np.ndarray, r: np.ndarray, *args: Any, **
        kwargs: Any) -> Any:
51              ...
```

The `Condition` class then, wraps a function from a set of labels in the dataset, to a numerical value. One may choose to combine multiple variables, or just one, as we have with the `"AGE"` variable in Figures 4 and 5. Please find an example initialisation in Figure 5. While one *can* inherit from the `Condition` class, we would advice to implement a custom function (as the `lambda` function we had), and provide it to the `Condition` object.

**Future goals & open-source** Our goal is for `AllSim` to be a benchmarking standard when evaluating ScRAPs. An important milestone for this, is to completely open-source our simulation. Doing so, allows other researchers to scrutinise, enhance, and discuss how we should move beyond what we can do today. Below, we provide two points we believe our community should discuss.

## E  Using `AllSim`

Here we provide the basic code, that will generate Figure 4. Naturally, the complete code is provided in the supplemental materials.

```
1      # load data
2      X, R, Y = custom_load_function(data)
3      organite = load_organite_model(location) # should load an implemented
        Inference object
4
5      # DENSITY LEARNING
6      # RESOURCES
7      bins_r = [30, 45, 60]
8      def condition_function_r(age):
9          return np.digitize(age, bins=bins_r).item()
10
11
12     condition_r = infer.Condition(
13         labels=['AGE'],
14         function=condition_function_r,
```

```python
15          options=len(bins_r) + 1
16      )
17
18      kde_r = infer.KDEDensity(condition=condition_r, K=condition_r.options
        )
19      kde_r.fit(R, one_hot_encoded=groups_r)
20
21
22      # RECIPIENTS
23      bins_x = [30, 45, 60]
24
25      def condition_function_x(bilir):
26          return np.digitize(bilir, bins=bins_x).item()
27
28      condition_x = infer.Condition(
29          labels=['AGE'],
30          function=condition_function_x,
31          options=len(bins_x) + 1
32      )
33
34      kde_x = infer.KDEDensity(condition=condition_x, K=condition_x.options
        )
35      kde_x.fit(X, one_hot_encoded=groups_x)
36
37
38      # BUILD THE SYSTEMS
39      resource_system, patient_system = dict(), dict()
40
41      def update_lam_0(t):
42          return (1 / (1+np.exp(-(t-450)/150))) * 3
43
44      def update_lam_1(t):
45          return (1 / (1+np.exp((t-350)/150))) * 2
46
47      def update_lam_2(t):
48          a = (1 / (1+np.exp((t-150)/150))) * 2
49          b = (1 / (1+np.exp(-(t-650)/100))) * 2
50          return (a + b)
51
52      def update_lam_3(t):
53          a = (1 / (1+np.exp(-(t-150)/150)))
54          b = (1 / (1+np.exp((t-650)/100)))
55          return (a + b)
56
57
58      resource_system[0] = infer.PoissonProcess(update_lam=update_lam_0)
59      resource_system[1] = infer.PoissonProcess(update_lam=update_lam_1)
60      resource_system[2] = infer.PoissonProcess(update_lam=update_lam_2)
61      resource_system[3] = infer.PoissonProcess(update_lam=update_lam_3)
62
63
64      patient_system[3] = infer.PoissonProcess(update_lam=update_lam_0)
65      patient_system[2] = infer.PoissonProcess(update_lam=update_lam_1)
66      patient_system[1] = infer.PoissonProcess(update_lam=update_lam_2)
67      patient_system[0] = infer.PoissonProcess(update_lam=update_lam_3)
68
69
70      resource_process = infer.PoissonSystem(
71          density=kde_r,
72          system=resource_system,
73          alpha=lambda t: 5,
74          normalize=True)
75
76      patient_process = infer.PoissonSystem(
77          density=kde_x,
```

```
78          system=patient_system,
79          alpha=lambda t: 7,
80          normalize=True)
81
82      organite.model.eval()
83
84      simulation = sim.Sim(
85          resource_system=resource_process,
86          patient_system=patient_process,
87          inference=organite
88      )
89
90      policy = MELD(
91          name='MELD', initial_waitlist=simulation._internal_waitlist, dm=
       dm
92      )
93
94      df = simulation.simulate(policy, T=1021)
```

### E.1  Getting started

To use AllSim, a user requires at least a dataset of the following type: $\mathcal{D} := \{(X_t, R_u, Y) : t, u \in \mathbb{N}_+\}$, with $t, u$ indicating the time of arrival. In principle, this is sufficient to start using AllSim already. In fact, this is exactly what we provided AllSim in our experiment in Figure 4. Let us elaborate:

AllSim requires to specify the following components:

1. A counterfactual model
2. The patient and resource arrival processes
3. The patient and resource densities

Components 1 and 3 are easily implemented using existing packages like econml (https://econml.azurewebsites.net) and scikit-learn (https://scikit-learn.org/stable/), respectively. They only require to use the model.fit API to be applied to the users' data:

```
1      counterf = econml.BaseCateEstimator()
2      counterf.fit(Y, R, X)
3
4
5      density_X, density_R = allsim.infer.KDEDensity(), allsim.infer.
       KDEDensity()
6      density_X.fit(X)
7      density_R.fit(R)
```

The only place where we need AllSim specifically is when we create the arrival processes, and the eventual simulation. Creating an arrival process on data, first requires us to regress the arrival probability on time:

```
1      from sklearn.preprocessing import PolynomialFeatures
2      from sklearn.linear_model import LinearRegression
3      from sklearn.pipeline import Pipeline
4
5      func_X = function(Pipeline([
6          ('poly', PolynomialFeatures(degree=6)),
7          ('linear', LinearRegression(fit_intercept=True))]))
8
9      # with amount() a function that returns how many of X arrived at t in
        D
10      func_X.fit(t, amount(X, t))
11
12      # with the arrival processes
```

```
13    patient_process = allsim.infer.PoissonProcess(func_X)
14    resource_process = allsim.infer.PoissonProcess(func_R)
15
16    patient_system = allsim.infer.PoissonSystem(density_X,
      patient_process)
17    resource_system = allsim.infer.PoissonSystem(density_R,
      resource_process)
```

The simulation is then created as:

```
1    simulation = asim.sim.Sim(
2        resource_system=resource_system,
3        patient_system=patient_system,
4        inference=counterf
5    )
```

The only thing left is to actually run the policy against our created simulation:

```
1    df = simulation.simulate(policy, T=100) # thats it!
```

## F   Social impact

It is very clear that machine learning has the potential to transform healthcare. Its success both in other domains and already within healthcare is very promising. ML-based policies have the potential to extend lives. However, as with almost any other machine learning method, there are risks associated with their deployment in a real healthcare setting. Before any (experimental) ML-based policy (or even non-ML-based policies for that matter) are to be deployed, they require thorough testing.

We believe `AllSim` will enable practitioners to leverage their data and evaluate their policies in a rigorous manner. `AllSim` fully recognises the difficulties associated with evaluating policies that diverge from the data-generating policy and aims to mitigate these difficulties by employing tried and tested methods from causality. Naturally, there are some caveats: `AllSim` inherits the potential assumptions made by the counterfactual method, furthermore, if real data is used, it is crucial it is at least *somewhat* related to the environment the tested policy will end up operating in. If for example, one aims to evaluate a policy on a domain that is completely unrelated, `AllSim`'s learned simulation will provide the tested policy with unrealistic resources, recipients, and arrivals.

Essentially, we believe AllSim may benefit (medical) society in the following (non-exhaustive) ways:

- We propose a shared evaluation benchmark for policies stemming from a wide variety of fields. Our focus is naturally ML, but fields such as OR and decision making can absolutely contribute. This allows the testing of policies for problems initially thought outside the scope of their original design.

- AllSim allows a shared platform for comparing results. Such a platform is missing in the allocation literature which until now had no formal way of comparing across different niche problems.

- AllSim allows much more thorough testing. Specifically, AllSim is able to stress test policies under extreme (unforseen) scenarios. This is an important part of testing which existing policies did not go through.

## G   AllSim in a vaccine distribution scenario

AllSim is a general purpose simulator which evaluates scarce resource allocation policies. While we mainly focus on organ-transplantation in our main text, we show in this section that AllSim is also applicable in other settings. To illustrate, we show how one can implement a vaccine distribution policy evaluation system in AllSim. This use-case will show how few adjustments one has to make with respect to the presented settings in our main text.

Some projects use AllSim Already:

- OrganITE and OrganSync (two organ allocation projects) are evaluated with an early iteration of AllSim [1, 2].
- OrganITE is currently being evaluated using this new version of AllSim, on different data and scenarios. This is a vital part of the process to get OrganITE in the hands of transplant centers.

Compared to the organ-allocation problem, in vaccine distribution, each resource is the same and they arrive in batches. Furthermore, the type of patient-in-need is also much broader (in fact, they cover the entire population). Yet, AllSim is perfectly capable of modelling this scenario given the following:

- Batch arrival simply requires a multiplier. For example, if the Poisson process samples a value of 2 on one day, we could simply interpret this as two batches of 1000 doses.
- As all vaccines are the same, we no longer require a density of resources as we required for organ allocation. This can be done by implementing a dummy-density that always returns 1 (or the amount of vaccine).
- A broader patient-type in AllSim is achieved by retraining the density of recipients over the entire population.

These implementation details are relatively simple to implement and easily done using AllSim's modular API.

While not necessarily a problem in vaccine distribution, recipient arrival in the ICU in a setting of infectious disease (such as COVID-19), is definitely different as compared to the organ-allocation setting. With organ-allocation, we can safely assume a Poisson process for recipient arrival as recipients enter the system independently. This is of course not true in an infectious disease scenario: one recipient arriving may indicate higher infection rate. As such, recipients *do not* arrive independently.

With the above, it is clear that we can no longer rely on a Poisson arrival process for recipients entering the system. Instead, to accurately model a situation of infectious disease, we recommend using a Hawkes process. To further illustrate, we include some code below showing exactly how one may go about including such a Hawkes process in AllSim.

```python
class HawkesProcess(PoissonProcess):
    def __init__(
            self,
            lam: float=.1,
            update_lam: Callable[[int], float]=lambda t: t,
            delta: float=.1,
            a: float=.2
        ):

        assert a >= 0, "a should be larger than or equal to 0"
        assert delta > 0, "delta should be larger than 0"

        super().__init__(lam, update_lam)

        self.a, self.delta = a, delta
        self._samples = []

    def get_lam_unnormalized(self, t: int) -> float:
        return self._baseline_lam + np.sum(
            self.a * self.beta * np.exp(-beta * (t - self._samples[
    self._samples < t])))

    def progress(self, t: int, neu: float=1) -> int:
        self.lam = neu * self.get_lam_unnormalized(t) # eqs. (5, 6)
        sample = np.random.poisson(lam=self.lam)
        self._samples.append(sample)
        return sample
```

Naturally, if recipient arrival is indeed dependent on previous arrivals, simply learning the arrival process (as we have for figure 3) should model such a self-exciting process automatically. Furthermore,

the above implementation is a simple linear univariate Hawkes process. We refer to `hawkeslib` (https://github.com/canerturkmen/hawkeslib) for implementations of more types of Hawkes processes.

## G.1 Online Learning

Consider the scenario where we wish to test a policy that learns continuously from newly presented users and resources (and of course outcomes). To allow for this, we present the `Policy.feedback(X, R, Y)` function. While it is not required to implement, the `simulation` calls this function whenever an outcome is simulated. This allows the policy to learn from its feedback.

The simple steps one needs to do are:

1. Implement a novel `Policy` subclass, which includes an `Inference` attribute (of the shape $\mathcal{X} \times \mathcal{R} \to \mathcal{Y}$.

2. Implement the `Policy.feedback` function to allow learning from the novel information (on outcomes) provided by the simulation

There are numerous algorithms which efficiently learn in the above framework [78, 79], which we recommend the future reader to consider.

## G.2 Low-level API example

```python
import allsim as asim

bins = [30, 45, 60]
f = lambda age: np.digitize(
        age, bins=bins).item()

condition = asim.infer.Condition(
    labels=["AGE"], function=f, options=4)

kde = asim.KDEDensity(
    condition=condition)
kde.fit(R) # R: res. pd.DataFrame

system = {
    0: asim.PoissonProcess(
        update_lam=regressors[0]),
    1: ...
}
poisson_system = asim.PoissonSystem(
    density=kde, system=system,
    alpha=alpha, normalize=True)

stream = DataFrame(
    columns=[*R_columns, "t"])
for t in T:
    sample = poisson_system(t)
    stream = stream.append(sample)
```

# H Counterfactual Inference

`AllSim` relies for a large part on counterfactual inference. As real-world data is collected under an active policy, testing an alternative policy would almost immediately diverge from the allocations made in the data. As such, we have to *infer* an outcome from a pair made by the tested policy. If the tested policy is indeed different from the active policy, then we are unlikely to find a comparable pair in the data. The above is a question most conisdered in research on counterfactual inference [69].

Of course there are many models that perform counterfactual inference. As such, we provide a brief overview of the type of models one may resort to. Naturally, this list is non-exhaustive, but may guide a user to a model fit for their use-case.

Broadly speaking, we recognise a few "meta-categories", or *meta-learners* [21, 80]. The are as follows:

- **T-Learner.** Simplest is to learn a model (such as a neural net or random forest) for each treatment. In the vaccine distribution case that would mean learning a model on recipients who haven't received a vaccine, and a different model on recipients who have received a vaccine.

- **S-Learner.** Contrasting the above, an S-Learning learns one model, where the treatment is considered part of the covariate set. This allows for a more flexible treatment, such as continuous [81] or multivariate treatments [1]. In our main-text we use OrganITE as a counterfactual model [2], which can be considered an S-Learner.

- **X-Learner.** Used specifically for estimating the treatment effect directly, an X-Learner first learns the outcome functions (such as the T-Learner), then imputes the dataset with the completed treatmet effect (or the estimated counterfactual outcome), and then learns the treatment effect with a third model directly on the completed samples. While useful for treatment effect estimating, performing counterfactual inference with an X-Learner would default back to either an S-Learner or T-Learner.

- **R-Learner.** An R-Learner, like the X-Learner, estimates the treatment effect directly [82]. Specifically, it learns the outcome functions as well as a propensity estimate. Using these two models, the R-Learner optimises a custom loss function based on the propensity, the outcome model, and a cross-validation setup.

- **DR-Learner.** The DR-Learner or *doubly robust*-learner is an iteration of the X-Learner [83]. Like the X-Learner the DR-learner first estimates the outcome models, then completes the dataset to learn a treatment effects model using standard supervised learning. Then the DR-Learner repeats this process using the treatment effects model from the first step.

While there are more meta-learners than what we reported above (e.g. U-Learner [82] or CW-Learner [84]), but they are much less adopted and unlike the S-Learner and T-Learner not fit for estimating the counterfactual outcomes as they, like the X-Learner, R-Learner, and DR-Learner, fit the effect function directly. We point the interested reader to the following papers: [21, 80, 82]; or to the following open-source libraries for various implementations: `causal-ml` (https://github.com/uber/causalml), or `econml` (https://github.com/microsoft/EconML).