# OpenReview forum: "AllSim: Simulating and Benchmarking Resource Allocation Policies in Multi-User Systems"
_NeurIPS.cc/2023/Track/Datasets_and_Benchmarks — NeurIPS 2023 Datasets and Benchmarks Poster_

### Official Review · Reviewer_uqdE · 2023-07-18

**Rating:** 3
**Confidence:** 5
**Clarity:** Yes, the writing is excellent.

**Strengths:**


(1) Overall, this paper is well-structured and easy to follow.

(2) The visualizations of key concepts are clear.

(3) The presentation of the paper is delectable.

**Additional Feedback:**

NA

**Correctness:**

It would be beneficial to conduct further testing and comparison of benchmark methods from the OR and RL fields.

**Documentation:**

The code and appendix require more effort to construct.

**Ethics:**

No.

**Limitations:**

Yes, the authors try to avoid potential negative societal by involving human verification in organ allocation.

**Opportunities For Improvement:**

(1) The authors overclaimed their contributions. It would be better to change the contribution to the organ allocation simulator.

(2) Online allocation problem is widely discussed in the operations research community. However, this paper only addresses organ allocation in a scenario where resources are received online. It fails to address a common form of resource allocation where users arrive online and consume resources from a resource pool (see the reference below). To provide a complete analysis, a thorough literature review is required along with the inclusion of additional cases (at least two more, such as those listed in Table 1) of online allocation in the code repository to support the stated contributions.

"Balseiro, S., Lu, H., & Mirrokni, V. (2020, November). Dual mirror descent for online allocation problems. In International Conference on Machine Learning (pp. 613-628). PMLR.".

(3) The appendix seems reused from their previous rebuttal without modification (e.g. line 723).

(4) In this paper, stochasticity refers to two factors: incoming resources and the user pool. From my understanding, operations research techniques can resolve the majority of online allocation problems with stochasticity primarily arising from incoming resources and minimal fluctuations in the user pool. Therefore, I suggest the authors provide an example of a significant change in the user pool that renders operations research methods ineffective.

(5) The code repository only includes the K-means clustering algorithm, which is a statistical method. It may be beneficial for the authors to also include some baseline methods for operations research and reinforcement learning.

**Relation To Prior Work:**

Yes.

**Summary And Contributions:**

In this article, the authors introduced a simulation tool called AllSim, designed to address the challenge of allocating organs (specifically, livers) online. They assert that AllSim is a versatile platform that can be used to simulate a wide range of resource allocation policies, making it useful for pre-deployment evaluations.

---

> ### Author Response · Authors · 2023-08-12
> **Thank you for your review**
>
> **_Thank you for reading our paper and providing comments which will help us improve the clarity and scope of our contribution. Below, we follow the structure of your review and respond to each comment seperately. We also list our actions taken for each comment._**
>
> Before responding point-by-point, we believe it may be instructive to  clarify that we are *not* proposing a novel allocation strategy. Instead, we show how to *evaluate* repeated resource allocation strategies. Naturally, this includes OR and RL policies and as such, are not competing, but _complementing_ research in these domains.
>
>
>
> **1. Contributions.** In our submission and in this response we argue that our contribution is applicable _beyond_ the organ allocation problem. That is because repeated resource allocation is a ubiquitous problem in many domains such as economics, business, education, etc.; transcending organ allocation alone. For multiple examples we refer to Tab. 1, Fig. 3, and Appendix G for a detailed example in vaccine distribution.
>
> Moreover, as we will show in our response to your point 2. (below), we show that papers such as Balseiro et al. (2020) (which is _not_ a paper on organ allocation) fit within this framework.
>
> _**Actions taken:** (i) We included a novel subsection 3.3 on the vaccine distribution setting in the main text. This shows a specific case for AllSim **outside** organ allocation. (ii) We include discussion on paper such as Balseiro et al. (2020) to further illustrate AllSim's generality._
>
>
>
>
>
>
> **2. Fixed resource allocation.** We truly appreciate the reviewer pointing us to different allocation scenarios. This allows us to further illustrate the generality of AllSim. In particular, the reviewer asks about scenarios which include:
> * online arrival of users (this is already modelled in the organ-allocation setting)
> * a fixed resource pool from which resources are taken and allocated to users
>
> Recall once more that the objective of AllSim is to _evaluate_ policies that solve for the scenario outlined above; AllSim's objective is _not_ to solve the scenario itself but to realistically simulate it.
>
>
>
> While AllSim supports unique and continuously arriving resources, that does not mean that AllSim _does not_ support a fixed set of generic resources such as those in Balseiro et al. (2020).
>
> In their paper, a policy needs to manage a resource pool ($B_i$). In AllSim, such a scenario is modelled as internal state in a policy. Simply inheriting the `Policy` class and keeping an attribute `Policy.B` which updates everytime the `Policy.get_x` function is called suffices to model the policy in Balseiro et al. (2020).
>
> Note that the arguments in `get_x` are generic in that they can accept $f_i$ and $b_i$ as specified in Balseiro et al. (2020).
>
>
> We believe that, by including the above example we show the power and generality of a simulator such as AllSim. We thank the reviewer for pointing us to this work which we will include in our paper.
>
> _**Actions taken:** (i) Include a new appendix which outlines alternative settings than the online user/resource setting described in the paper. (ii) Extend our new sec. 3.3 with a fixed resource implementation using Balseiro et al. (2020)._
>
>
>
> **3. Appendix.** This was an oversight which is now fixed. Thank you for pointing this out.
>
> _**Actions taken:** Fixed typo in our Appendix._
>
>
> **4. OR methods in AllSim.** We wish to emphasise that our paper doesn't express an opinion on when one should default to OR, RL, or alternative allocation policy designs. In fact, AllSim is _agnostic_ to what area of literature a certain policy is stemming from. The focus is on building a testing environment for _any_ policy type that solves the repeated resource allocation problem, including techniques from the OR community.
>
> We will mention in our conclusion that we hope that the OR community will use AllSim to test their policies, and perhaps extend AllSim based on their unique requirements.
>
> We thank the reviewer for their great suggestion!
>
> _**Actions taken:** (i) Move OR paragaphs of App. B.1 to main text. (ii) Explain in our main text that OR strategies (and RL strategies) can be evaluated using AllSim. (iii) Extend the conclusion as explained above._
>
>
>
> **5. OR and RL allocation.** K-Means is part of the _simulation_, not the _policy_. K-Means is a way to automatically build subgroups used to learn conditional densities. These densities are later used to sample users and resources within the simulated environment.
>
> Naturally, there are other ways to build such conditional densities, but too our knowledge, OR and RL methods are not one of them. Do not that we other density designs in AllSim's API by allowing users to implement their own densities using the abstract `Density` class.
>
>
>
> _**We sincerely thank the reviewer for their helpful review of our paper. We believe their comments helped us clarify the usefulness of AllSim, and its potential for adoption in the OR community.**_

---

### Official Review · Reviewer_tL4u · 2023-07-20

**Rating:** 6
**Confidence:** 3
**Correctness:** Yes.

**Strengths:**

This paper provides the means to perform standardised evaluation of repeated resource allocation policies in non-steady-state environments. AllSim’s generality and modularity allows for sensible adoption in a wide range of application areas. Furthermore, having standardised evaluation will encourage research in this very important and impactful domain spanning many application areas.

**Additional Feedback:**

No.

**Clarity:**

The layout structure of the article appears to be clear, and readers can find relevant core content while reading the text.

**Documentation:**

Yes.

**Ethics:**

No.

**Limitations:**

I believe that the author has fully considered limitations and potential negative social impacts, and provided corresponding solutions, not only in terms of policy assistance but also ethical and moral choices. At the same time, it is suggested that the author can provide more positive impacts of the system on other aspects of society beyond these two aspects.

**Opportunities For Improvement:**

N/A

**Relation To Prior Work:**

N/A

**Summary And Contributions:**

1.	This paper released a general-purpose open-source framework for performing data-driven simulation of scarce resource allocation policies for pre-deployment evaluation.
2.	AllSim uses modular environment mechanisms to capture a range of environment conditions, allowing users to further configure parameters for stress testing and sensitivity analysis.
3.	Compared to existing work, we believe this simulation framework takes a step towards more methodical evaluation of scarce resource allocation policies.

---

> ### Author Response · Authors · 2023-08-12
> **Thank you for your review**
>
> **_Thank you for reading our paper and providing comments which will help us improve the clarity and scope of our contribution. Below, we follow the structure of your review and respond to each comment seperately. We also list our actions taken for each comment._**
>
>
> **Societal impact.** We truly believe in AllSim's potential for positive impact in society. Here are some possible ways in which AllSim may express this positive impact:
> * We propose a shared evaluation benchmark for policies stemming from a wide variety of fields. Our focus is naturally ML, but fields such as OR and decision making can absolutely contribute. This allows the testing of policies for problems initially thought _outside_ the scope of their original design.
> * AllSim allows a shared platform for comparing results. Such a platform is missing in the allocation literature which until now had no formal way of comparing across different niche problems.
> * AllSim allows much more thorough testing. Specifically, AllSim is able to stress test policies under extreme (unforseen) scenarios. This is an important part of testing which existing policies did not go through.
>
> Naturally there are more areas where we may expect positive impact. We believe, based on the reviewer's comment, that a future "testimonials" page on our GitHub page would be an encouraging (and expanding) collection of areas of positive societal impact.
>
> _**Actions taken:** (i) We expanded Appendix F with above examples. (ii) We are preparing a "testimonials" page within our online (open-sourced) code repository and documentation._
>
>
> _**We sinscerely thank the reviewer for their helpful review of our paper. We believe their comments greatly improve our submission! We would love to engage further with the reviewer and look forward to answer any remaining questions.**_

---

### Official Review · Reviewer_XpaH · 2023-07-21
**A repeated resource allocation policies evaluation method based on counterfactual inference.**

**Rating:** 6
**Confidence:** 4
**Correctness:** No incorrectnesses were found.

**Strengths:**

1.Reasonable parameterized modeling of the arrival process: On the one hand, different user (resource) types can be decomposed and fine-tuned, while on the other hand, it also allows to simulate a user-specific drift. This adapts to more scenarios in this field and provides guidance for future researchers.

2.The generation strategy based on counterfactual inference generates simulation results by modeling causal relationships rather than correlation relationships, which helps researchers in the field generate more meaningful and available data.

**Additional Feedback:**

It seems that there is a lack of visual comparison of different counterfactual inference methods on the final results.

**Clarity:**

The writing of the paper needs improvement. For example, a typed wrong within “We believe this is due to the lack of proper evaluation tools as, too our knowledge, ” in the introduction.

**Documentation:**

The datasets used in the paper are adequately documented.

**Ethics:**

The resource releases on datasets that are widely used in the community. So I do not see any ethical concerns.

**Limitations:**

The availability of the data generated in this paper strongly depends on the availability of the counterfactual inference method itself, which may be affected by factors such as confounding factors. The author did not point out the limitations of this aspect.

**Opportunities For Improvement:**

The main contribution of this paper is how to use counterfactual inference to generate corresponding results for different decisions, but we can often only use relevant relationships for modeling in real applications. Although there is a lack of explanatory power, in certain situations such as the presence of confounding factors, it isn't easy to establish accurate causal relationships, and relying solely on correlation can yield relatively accurate results.

Therefore, I suggest incorporating relevance-based inference as a parallel technique to counterfactual inference into the framework.

**Relation To Prior Work:**

The paper is adequately contextualized with respect to prior work.

**Summary And Contributions:**

The author provides a usable framework based on counterfactual reasoning to evaluate different repeated resource allocation policies. Specifically, this paper provides its insights on three challenges that have not yet been fully studied and they are

1.The arrival process of resources and users should be influenced by multiple complex effects (including complex interactive utility).

2.Users, resources, and decisions should be an overall system, and when a certain factor changes, it will change as a whole. Therefore, historical data based on specific decisions will introduce bias.

3.In multi-user problems, the impact of each decision will not be limited to resource recipients but should consider all users.

In response to the above issues, this paper divides the evaluation framework into multiple components, models the decision-making process reasonably through different parameterization methods, and provides unbiased generated data based on counterfactual inference methods for reasonable comparison and evaluation.

---

> ### Author Response · Authors · 2023-08-12
> **Thank you for your review**
>
> **_Thank you for reading our paper and providing comments which will help us improve the clarity and scope of our contribution. Below, we follow the structure of your review and respond to each comment seperately. We also list our actions taken for each comment._**
>
>
> **Associative models in favour of counterfactuals.** While our paper _does_ focus mostly on counterfactual models, the reviewer is correct: counterfactual models are sometimes inferior to simple associative models. Ultimately, AllSim _does_ allow for non-causal methods, despite the API suggesting otherwise. Essentially, a counterfactual model infers $Y$ from ${X, R}$, which shares its "API" with an associative model.
>
> To encourage a more wide set of inference models (beyond counterfactual models), we subclassed the existing `Inference` class with an example non-counterfactual model: `MLP`, which is a simple MLP learning naively from the provided dataset.
>
> Naturaly, more involved and exciting inference techniques than `MLP` are possible within the `Inference` API, but we leave those for the practiioner themselves to implement. In case these novel implementations are issued in a pull-request (of our future GitHub page), they will be reviewed and consequentially accepted into the AllSim codebase.
>
> _**Actions taken:** Implement a novel `MLP` class which inherets from the abstract `Inference` class as an alternative to counterfactual methods._
>
> _**We sinscerely thank the reviewer for their helpful review of our paper. We believe their comments greatly improve our submission! We would love to engage further with the reviewer and look forward to answer any remaining questions.**_

---

> > ### Comment · Reviewer_XpaH · 2023-08-29
> >
> > Thanks to the authors for their efforts in addressing my concerns.
> >
> > 1.The author's response basically addresses my main concern, which is to provide an alternative method for counterfactual inference as a supplement. Although I did not actually perform the operation, the author has clearly described the entire process.
> >
> > 2.It seems that the author has not made further detailed modifications to the paper, and the errors I mentioned earlier (such as typos) still seem to have reservations. If the author has already provided a modified version, please provide the original version and indicate the modified part.

---

> > > ### Author Response · Authors · 2023-08-30
> > > **Thank you for your engagement!**
> > >
> > > _Dear reviewer XpaH, we want to thank you for engaging with us! This is truly appreciated._
> > >
> > > 1. We are happy to hear that our clarifications have addressed your main concern.
> > >
> > > 2. We have uploaded a new version of our paper. We apologise for this inconvenience. With respect to your specific typo, we changed our wording to:
> > >
> > > > We believe the reason is the lack of proper evaluation tools; to our knowledge, there only exist tools that: [...]
> > >
> > > Hopefully, our revised manuscript resolves your remaining reservations. If not, please do not hesitate to ask further questions!
> > >
> > > Best,
> > >
> > > Authors of 817

---

### Official Review · Reviewer_pbmH · 2023-07-22
**Good paper**

**Rating:** 8
**Confidence:** 4
**Correctness:** Yes
**Clarity:** Yes

**Strengths:**

I believe this paper is quite meaningful for the resource allocation community. I really like that in the appendix, the authors listed examples from different multi-user problems in different areas. I looked into the code and the appendix helped me understand the structure quite easily. As the dataset and benchmark paper, it's important for users to follow the work and achieve easy reproduction.

**Additional Feedback:**

NA

**Documentation:**

Yes

**Ethics:**

No ethics issues.

**Limitations:**

The authors can consider discussing the overhead of the system and evaluate their system in a more comprehensive way. The evaluation for each example seems a bit weak to me. The author did mention in the paper that they only display some of the results, but for the paper, it seems that the authors should at least have one completed evaluation for one example.

**Opportunities For Improvement:**

I think one improvement may be to put everything into a nice GitHub page or build an online doc. This is just a suggestion. Most of the researchers did it when they are trying to help users learn and reproduce their work. For only paper submission/review, I think this appendix and code in zip file is enough, but for the community, the authors should consider the open source option.

**Relation To Prior Work:**

Yes

**Summary And Contributions:**

This paper presents a benchmarking environment, namely ALLSim, which provides a realistic simulation of the effect and usefulness of resource allocation policies in systems where users are vying for limited resources. This is the key contribution of the paper. The ALLSim system allows for a more organized assessment of policies managing scarce resource allocation, improving upon existing benchmarking techniques for such methods.

---

> ### Author Response · Authors · 2023-08-12
> **Thank you for your review**
>
> **_Thank you for reading our paper and providing comments which will help us improve the clarity and scope of our contribution. Below, we follow the structure of your review and respond to each comment seperately. We also list our actions taken for each comment._**
>
>
> **GitHub & Open-source.** We consider open-sourcing our code/project on GitHub to be **enourmously** important. Hence, we completely agree with the reviewer that this is an important next step to:
> * Encourage adoption
> * Allow community driven API changes
> * Receive feedback on where we move AllSim next
>
> We welcome and look forward to these community driven changes and feedback (either clinical or CS) and GitHub is (in our view) the best way to do this.
>
>
> **Additonal results.** Since building AllSim, we have used it in several clinical projects on organ-allocation. It is our ambition to include these projects as examples in our documentation which accompanies the GitHub mentioned above.
>
> _**Actions taken:** We have expanded Appendix G with the projects we can already share using AllSim. Future (and in projects in submission) will be included in our GitHub documentation._
>
>
>
> _**We sinscerely thank the reviewer for their helpful review of our paper. We believe their comments greatly improve our submission! We would love to engage further with the reviewer and look forward to answer any remaining questions.**_

---

### Official Review · Reviewer_zixr · 2023-07-24
**Impact problem and promising results**

**Rating:** 7
**Confidence:** 3

**Strengths:**

1. AllSim have broad relevance across multiple high-impact problems
2. AllSim API includes several base utilities to model diverse distributions, state-of-the-art methods for counterfactual inference and easy integration with numpy and pandas. This can allow for quick data ingestion and post-hoc analysis.

**Additional Feedback:**

The code snippets in the paper are helpful but would also be useful if you can share the github repo.

**Clarity:**

The paper is well written and easy to follow. The appendix is very well organized and easy to follow.

**Correctness:**

Evaluation metrics and experiment design details are included in the main paper. Multiple scenarios are simulated. Correctness is tough to validate in such a scenario such the contribution is a package without looking at github repo.

**Documentation:**

Code snippets are included in the paper and supplementary. But, access to the github repo would be helpful to test how the system works.

**Ethics:**

No ethical concerns.

**Limitations:**

Authors have briefly mentioned current limitations in the work (eg: support continuous time settings). However, there is no such dedicated section in the paper.

**Opportunities For Improvement:**

Comments for the paper:
1. Slight inconsistencies between the definition and the design of API. This is slightly confusing and not-very intuitive. Took me a while to understand code snippet in figure 5 and still unclear on how it connects to code in section 3.2. Maybe have a visual diagram for the API schema.
- After reading the definition, I was expecting the API to be have high-level wrappers like: add_dataset(), model_process(); evaluate_policy(), get_counterfactual().
- High-level wrappers can improve readability of code and use of API. Fot instance, could use a model_process wrapper for user and resource processes instead of directly exposing KDEDensity and PoissonSystem functions. This will make it more intuitive, especially when users likely to be domain experts.
-Question: Does asim.simulation.simulate execute the evaluation with a given policy? Is there a different evaluation API? I am slightly confused about this nomenclature.

2. While policies can be evaluated pre-deployment in AllSim, would be cool to support ability to also learn policies online. This could be done based on constraints based on changes in environment variables - users, arrival times, resources.
- Is it possible with the current API design given that it not auto-diff code? Can we evaluate learned policies (eg: from pytorch) in this systems? Would you want to integrate methods in causal representation learning etc and support high-dim input datasets which can be processed using DNN?

3. Providing more experimental analysis and access to code will make it a stronger submission.

General Suggestion:
- Develop data loaders to multiple real-world sources of relevance. Will make it easier for domain experts and users to onboard onto the framework.
- Building an interactive visualization toolkit to run counterfactual inference (for pre-deployment evaluation) from a UI would make this project more impactful, given users are likely to be non-computer scientist decision makers.

**Relation To Prior Work:**

No, this is only briefly discussed. Would like to see a broader discussion of similar frameworks in the space. The paper should summarize pros and cons of different frameworks in a table or section. What are the top-5 closest frameworks to this project? How are they different?

**Summary And Contributions:**

The paper introduce AllSim - a framework for data-driven simulation of scarce resource allocation policies for pre-deployment evaluation. The problem is high-impact and relevant across multiple domains from patient-organ matching transport to vaccine allocation to headhunting. First, AllSim ingests a dataset of users, resources and outcomes. Second, AllSim uses D to identify a distribution over resources, users and arrival times and specifies a utility function u. Third, a decision maker can now leverge AllSim for counterfactual inference to design variations of the allocation policy. Often, this can be used to generate a new batched dataset D' (shipped as a pandas df) that can be used for downstream analysis. The paper describes the AllSim API through a case-study of organ transplanation on a real dataset (and vaccine allocation in the appendix)

---

> ### Author Response · Authors · 2023-08-12
> **Thank you for your review**
>
> **_Thank you for reading our paper and providing comments which will help us improve the clarity and scope of our contribution. Below, we follow the structure of your review and respond to each comment seperately. We also list our actions taken for each comment._**
>
> **1. API Design.** We thank the reviewer for these suggestions! The code snippet in fig. 5 can indeed by simplified by including a more high level API, which in turn will result in wider adoption.
>
> We propose the following alternative:
> ```python
> import allsim as asim
>
> simulation = asim.init(**kwargs)        # Kwargs could include:
>                                         # - arrival process types (defined in API)
>                                         # - counterfactual model type (defined in API)
>
> simulation.add_dataset(X, R, Y)         # Adding a dataset should automatically model arrival processes
>
> df = simulation.evaluate(policy, T)
>
> ```
>
> In the design above, we would allow inheritance from the `Simulation` class, which in turn allows implementing hooks in the simulation life-cycle (as per fig. 6 in App. A). A comparable example to such an API design in `pytorch-lightning` which has a basic training loop, but more complicated models are still possible through custom implementation of hooks in their life-cycle.
>
> _**Actions taken:** (i) We adjusted the example in our main text with the above to illustrate a general high-level implementation. (ii) We adjusted our API design to allow for more high-level interaction. (iii) We included a more detailed example in Appendix G which illustrates the steps in our old Fig. 5._
>
>
> **2. Online learning.** Indeed! We agree with the reviewer that online learning of policies is a very high potential use-case of AllSim. In fact, the current API already allows us to do this!
>
> In principle, there are very few constraints on the `Policy` class, and the only real interaction between the `Policy` and the `Simulation` are very comparable to what one would expect from a `gym` environment (Brockman et al., 2016).
>
> As such, the `Policy` class can absolutely use a DNN as internal approximator, while also updating using auto-diff. In particular, one needs to implement a custom `Policy.feedback` function, which can be called in a callback when inheriting from `Simulation`.
>
> Of course, we find the above somewhat cumbersome and agree with the reviewer that such use-cases can make adoption of AllSim more interesting. Hence, we will make the `Policy.feedback(X, Y)` function part of the current API. This will automatically be called in the simulation life-cycle. A new policy can simply rely on the `feedback` function, but there is no mandate to do so.
>
> _**Actions taken:** (i) We extend our API to include `Policy.feedback`. (ii) We extend Appendix G to give an example of an online learning policy._
>
>
> **3. Experiments and code.** A _major_ part of AllSim is open-sourcing our complete code base on github. To encourage adoption, many examples and documentation is **key**, deserves (and will receive) our full attention.
>
>
>
> **General suggestions:**
> * With our new API design (`add_dataset`), we propose to also include a `load_dataset` much like the standard ML-frameworks. Our case will be a little more involved as many datasets (such as UNOS) are publicly available _only upon request_. Sadly, such datasets are currently not accessable using API keys. However, we are hopeful that by introducing a `load_dataset` API in combination with wide adoption, that a registry of online datasets is possible in the future.
> * We are actually actively working on such a visualisation! As you suggest, many of our clinical (non CS) partners are asking for a more visual way to interact with AllSim (and the tested policies in general).
>
>
> _**We sinscerely thank the reviewer for their helpful review of our paper. We believe their comments greatly improve our submission! We would love to engage further with the reviewer and look forward to answer any remaining questions.**_

---

### Decision · Program_Chairs · 2023-09-22

**Decision:**

Accept (Poster)

**Comment:**

Overall, reviewers found this paper interesting and focuses on a timely and important problem.

We do encourage the authors to improve this paper by fixing reviewers' concern, especially @uqdE. It would be great if the contributions can be scoped more clearly and other issues fixed in the final version.